# Unsupervised Bayesian Ising Approximation for decoding neural activity and other biological dictionaries

**Damián G Hernández[1,2]\*, Samuel J Sober[3]\*, Ilya Nemenman[2,3,4]\***

[1]Department of Medical Physics, Centro Atómico Bariloche and Instituto Balseiro, Bariloche, Argentina; [2]Department of Physics, Emory University, Atlanta, United States; [3]Department of Biology, Emory University, Atlanta, United States; [4]Initiative in Theory and Modeling of Living Systems, Atlanta, United States

**Abstract** The problem of deciphering how low-level patterns (action potentials in the brain, amino acids in a protein, etc.) drive high-level biological features (sensorimotor behavior, enzymatic function) represents the central challenge of quantitative biology. The lack of general methods for doing so from the size of datasets that can be collected experimentally severely limits our understanding of the biological world. For example, in neuroscience, some sensory and motor codes have been shown to consist of precisely timed multi-spike patterns. However, the combinatorial complexity of such pattern codes have precluded development of methods for their comprehensive analysis. Thus, just as it is hard to predict a protein's function based on its sequence, we still do not understand how to accurately predict an organism's behavior based on neural activity. Here, we introduce the unsupervised Bayesian Ising Approximation (uBIA) for solving this class of problems. We demonstrate its utility in an application to neural data, detecting precisely timed spike patterns that code for specific motor behaviors in a songbird vocal system. In data recorded during singing from neurons in a vocal control region, our method detects such codewords with an arbitrary number of spikes, does so from small data sets, and accounts for dependencies in occurrences of codewords. Detecting such comprehensive motor control dictionaries can improve our understanding of skilled motor control and the neural bases of sensorimotor learning in animals. To further illustrate the utility of uBIA, we used it to identify the distinct sets of activity patterns that encode vocal motor exploration versus typical song production. Crucially, our method can be used not only for analysis of neural systems, but also for understanding the structure of correlations in other biological and nonbiological datasets.

**\*For correspondence:**
damian.g.h.l@gmail.com (DGH);
samuel.j.sober@emory.edu (SJS);
ilya.nemenman@emory.edu (IN)

**Competing interest:** The authors declare that no competing interests exist.

## Editor's evaluation

This work introduces a new unsupervised Bayesian method for identifying important patterns in neural population responses. The method offers improvements relative methods that do not assume correlations in neural responses, and is likely to also outperform methods that take into account pairwise correlations in neural responses.

## Introduction

One of the goals of modern high-throughput biology is to generate predictive models of interaction networks, from interactions among individual biological molecules (*Marks et al., 2011*) to the encoding of information by networks of neurons in the brain (*Schneidman et al., 2006*). To make predictions about activity across networks requires one to accurately approximate—or build a *model*

**Figure 1.** The dictionary reconstruction problem. In many biological systems, such as understanding the neural code, identifying protein-DNA binding sites, or predicting 3-D protein structures, we need to infer dictionaries — the sets of statistically over- or under-represented features in the datasets (relative to some null model), which we refer to as words in the dictionary. To do so, we represent the data as a matrix of binary activities of $N$ biological units (spike/no spike, presence/absence of a mutation, etc.), and view $M$ different experimental instantiations as samples from an underlying stationary probability distribution. We then use the uBIA method to identify the significant words in the data. Specifically, uBIA systematically searches for combinatorial activity patterns that are over- or under-represented compared to their expectation given the marginal activity of the individual units. If multiple similar patterns can (partially) explain the same statistical regularities in the data, they compete with each other for importance, resulting in an irreducible dictionary of significant codewords. In different biological problems, such dictionaries can represent neural control words, DNA binding motifs, or conserved patterns of amino acids that must be neighbors in the 3-D protein structure.

of—their joint probability distribution, such as the distribution of joint firing patterns in neural populations or the distribution of co-occurring mutations in proteins of the same family. To successfully generalize and to improve interpretability, models should contain as few as possible terms. Thus constructing a model requires detecting *relevant* features in the data: namely, the smallest possible set of spike patterns or nucleotide sequences that capture the most correlations among the network components. By analogy with human languages, where words are strongly correlated co-occurring combinations of letters, we refer to the problem of detecting features that succinctly describe correlations in a data set as the problem of *dictionary reconstruction*, as schematized in *Figure 1*. It is the first step towards building a model of the underlying data, but it is substantially different (and potentially simpler) than the latter: detecting which features are relevant is not the same as quantifying how they matter in detail.

In recent years, the problem of dictionary reconstruction has been addressed under different names for a variety of biological contexts (*Natale et al., 2018*) including gene expression networks (*Margolin et al., 2006*; *Lezon et al., 2006*), protein structure, protein-protein interactions (*Marks et al., 2011*; *Morcos et al., 2011*; *Bitbol et al., 2016*; *Halabi et al., 2009*), the structure of regulatory DNA (*Otwinowski and Nemenman, 2013*), distribution of antibodies and pathogenic sequences (*Mora et al., 2010*; *Ferguson et al., 2013*), species abundance (*Tikhonov et al., 2015*), and collective behaviors (*Bialek et al., 2012*; *Couzin and Krause, 2003*; *Lukeman et al., 2010*; *Kelley and Ouellette, 2013*; *Pérez-Escudero and de Polavieja, 2011*). The efforts to identify interactions in neural activity have been particularly plentiful (*Stevens and Zador, 1995*; *Schneidman et al., 2006*; *Pillow et al., 2008*; *Bassett and Sporns, 2017*; *Williams, 2019*). The diversity of biological applications notwithstanding, most of these attempts have relied on similar mathematical constructs, and

most have suffered from the same limitations. First, unlike in classical statistics and traditional quantitative model building, where the number of observations, $M$, usually vastly exceeds the number of unknowns to be estimated, $K$, $K/M \ll 1$, modern biological data often has $M \gg 1$, but also $K/M \gg 1$. Indeed, because of network features such as protein allostery, recurrent connections within neural populations, and coupling to global stimuli, biological systems are rarely limited to local interactions only (*Schwab et al., 2014*; *Merchan and Nemenman, 2016*; *Nemenman, 2017*), so that the number of pairwise interactions among $N$ variables is $K \sim N^2$, and the number of all higher order interactions among them is (that is, interactions that involve more than two network variables at the same time) is $K \sim 2^N$. Put differently, words in biological dictionaries can be of an arbitrary length, and spelling rules may involve many letters simultaneously, some of which are far away from each other. Because of this, reconstruction of biological dictionaries from data sets of realistic sizes requires assumptions and simplifications about the structure of possible biological correlations, and will not be possible by brute force. The second problem is that, as in human languages, biological dictionaries have redundancies: there are synonyms and words that share roots. For example, a set of gene expressions may be correlated not because the genes interact directly, but because they are related to some other genes that do. Similarly, a certain pattern of neural activity may be statistically over- or under-represented (relative to a null model) not on its own, but because it is a subset or a superset of another, more important, pattern. Identifying *irreducible words*—the root forms of biological dictionaries—is therefore harder than detecting all correlations while also being crucial to fully understanding biological systems. Together, these complications make it impossible to use standard methods for reconstructing combinatorially complex dictionaries from datasets of realistic sizes.

In this work, we propose a new method for reconstructing complex biological dictionaries from relatively small datasets, as few as $M \sim 10^2 \ldots 10^3$ samples of the joint activity and test it on neural data from songbirds. We focus on the regime $M \gg N$, which means $N$ of a few tens for our datasets. While small compared to some of the largest high throughput biological datasets, this regime is relevant in many biological contexts, and especially in studies of motor systems, where recording from multiple single motor units is hard. Crucially, the method imposes no limitation on the structure of the words that can enter the dictionary — neither their length nor their rules of spelling — beyond the obvious limitation that (i) words that do not happen in the data cannot be detected, and (ii) that data contain few samples of many words, rather than of just a few that repeat many times. Additionally, we address the problem of irreducibility, making the inferred dictionaries compact, non-redundant, and easier to comprehend. The main realization that allows this progress is that instead of building an explicit model of the entire joint probability distribution of a system's states and hence answering *how* specific significant words matter, we can focus on a more restricted, and thus possibly simpler, question: *which* specific words contribute to the dictionary. In other words, unlike many other methods, we do not build an explicit model of the underlying probability distribution, which would allow us to 'decode' the meaning of the data, but only detect features that can be used in such models later. We do this using the language of Bayesian inference and statistical mechanics by developing an unsupervised version of the Bayesian Ising Approximation (*Fisher and Mehta, 2015*) and by merging it with the *reliable interactions* model (*Ganmor et al., 2011*).

We believe that the approach we develop is fully general and will allow analysis of diverse datasets with realistic size requirements, and with few assumptions. However, to illustrate the capabilities of the approach, we present it here in the context of a specific biological system: recordings from single neurons in brain area RA (the robust nucleus of the arcopallium) in the Bengalese finch, a songbird. Neurons communicate with each other using patterns of action potentials (spikes), which encode sensory information and motor commands, and hence behavior. Reconstructing the neural dictionary, and specifically detecting irreducible patterns of neural activity that correlate with (or 'encode') sensory stimuli or motor behaviors — which we hereafter call *codewords* — has been a key problem in computational neuroscience for decades (*Stevens and Zador, 1995*). It is now known that in both sensory (*Berry et al., 1997*; *Strong et al., 1998*; *Reinagel and Reid, 2000*; *Arabzadeh et al., 2006*; *Rokem et al., 2006*; *Nemenman et al., 2008*; *Lawhern et al., 2011*; *Fairhall et al., 2012*) and motor systems (*Tang et al., 2014*; *Srivastava et al., 2017*; *Sober et al., 2018*; *Putney et al., 2019*) the timing of neural action potentials (spikes) in multispike patterns, down to millisecond resolution, can contribute to the encoding of sensory or motor information. Such dictionaries that involve long sequences of neural activities (or incorporate multiple neurons) at high temporal resolution are both

more complex and more likely to be severely undersampled. Specifically, even though the songbird datasets considered here are large by neurophysiological standards, they are too small to build their statistical models, which is the goal of most existing analysis approaches. This motivates the general inference problem we address here.

The power of our approach is illustrated by discoveries in the analysis of the songbird vocal motor code. Specifically, while it is known that various features of the complex vocal behaviors are encoded by millisecond-scale firing patterns (*Tang et al., 2014*), here we identify which specific patterns most strongly predict behavioral variations. Further, we show that dictionaries of individual neurons are rather large and quite variable, so that neurons speak different languages, which nonetheless share some universal features. Intriguingly, we detect that codewords that predict large, exploratory deviations in vocal acoustics are statistically different from those that predict typical behaviors. Collectively, these findings pave the way for development of future theories of the structure of these dictionaries, of how they are formed during development, how they adapt to different contexts, and how motor biophysics translates them into specific movements. More importantly, the development of this method and its successful application to neural and behavioral data from songbirds suggests its utility in other biological domains, where reconstruction of feature dictionaries is equally important, and where new discoveries are waiting to be made.

## Results

### The dictionary reconstruction problem

We formalize the dictionary reconstruction problem as follows. An experimental dataset consists of $M$ samples of a vector of binary variables of length $N$, $\vec{\sigma} = \{\sigma_i\}_0^N$, which we call *letters* (or *spins*, by analogy with statistical physics). These samples are assumed to be taken from a stationary joint probability distribution $P(\vec{\sigma})$, but the distribution is unknown. From the frequency of occurrence of various combinations of σs in the dataset, we need to detect *words* in the model defined by $P(\vec{\sigma})$, namely those patterns of σs that are significantly over- or under-represented (statistically anomalous) compared to some null model. While different null models are possible, a common choice is the product of marginal distributions $P_{\text{null}} = \prod_{i=1}^N P(\sigma_i)$. In this case, words are defined as *correlated* patterns of binary letters. Additionally, to be a part of the dictionary, a word must be *irreducible*. That is, it must be statistically anomalous not only with respect to the null model, but also with respect to its own various parts. For example, a word $\sigma_1\sigma_2\sigma_3$ all equal to 1 is a word in a dictionary only if it is statistically anomalous with respect to its frequency expected from frequencies of $\sigma_1\sigma_2$, $\sigma_1\sigma_3$, $\sigma_2\sigma_3$, and also from frequencies of $\sigma_1\sigma_2\sigma_3\sigma_4$, $\sigma_1\sigma_2\sigma_3\sigma_5$, $\sigma_1\sigma_2\sigma_3\sigma_4\sigma_5$, and so on, eventually accounting for all other words that include any combination of letters $\sigma_1$, $\sigma_2$, and $\sigma_3$. In principle, such statistical over- or under-representation of a word has a precise mathematical definition, based on comparing the entropy of the maximum entropy distribution constrained by frequencies of all other words in the distribution to that constrained additionally by the word itself (*Margolin et al., 2010*). For example, three anomalous overrepresented words are shown in the top right panel of *Figure 1*. Similarly, an example of anomalous underrepresentaion is shown in light blue in *Figure 1*, right middle panel, in which the word 1.1 is underrepresented relative to the frequency expected from the (marginal) frequencies of 1·· and ··1. In practice, performing these many comparisons is impossible for all but the simplest cases. Even approximate analyses, aiming to prove that a method results in irreducible dictionaries under some specific assumptions, have not been very successful to date. As a result, typical methods for dictionary reconstruction are assessed in their ability to build irreducible dictionaries based on heuristics, such as having a low probability of including overlapping words. We will follow the same route here.

### Songbird neural motor code as a dictionary reconstruction problem

Owing to their complex and tightly regulated vocal behavior and experimentally accessible nervous system, songbirds provide an ideal model system for investigating the neural dictionaries underlying complex motor behaviors (*Doupe and Kuhl, 1999*; *Kuebrich and Sober, 2015*). We recorded from individual neurons in the motor cortical area RA of Bengalese finches during spontaneous singing and quantified the acoustic 'features' of song, specifically the fundamental frequency (which we will refer to as 'pitch'), amplitude, and spectral entropy of individual vocal gestures, or 'syllables', as described previously (*Sober et al., 2008*; *Tang et al., 2014*; *Wohlgemuth et al., 2010*). The data sets

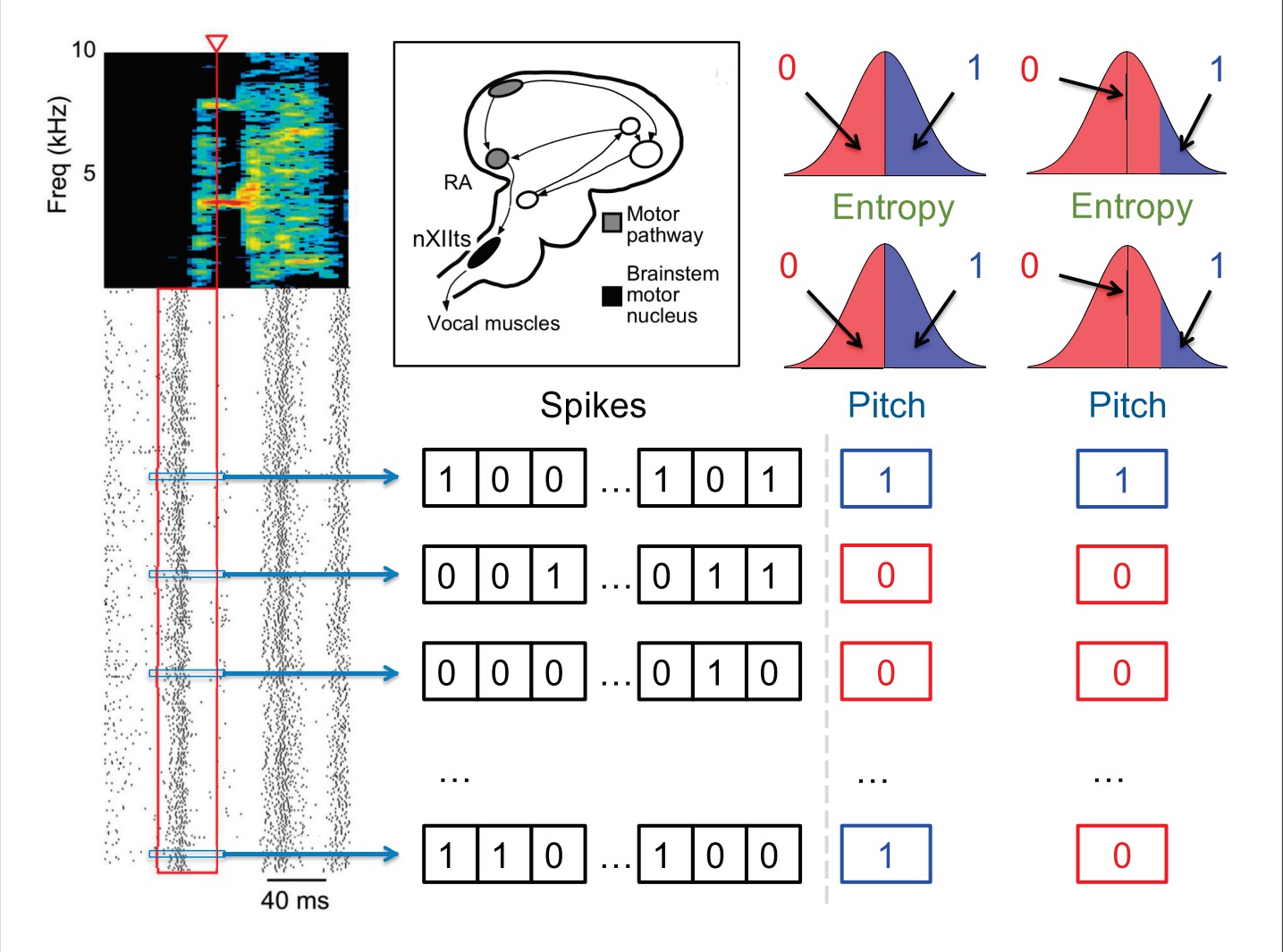

**Figure 2.** Quantification of the neural activity and the behavior. A spectrogram of a single song syllable in top-left corner shows the acoustic power (color scale) at different frequencies as a function of time. Each tick mark (bottom-left) represents one spike and each row represents one instantiation of the syllable. We analyze spikes produced in a 40ms premotor window (red box) prior to the time when acoustic features were measured (red arrowhead). These spikes were binarized as 0 (no spike) or 1 (at least one spike) in 2ms bins, totaling 20 time bins. The different acoustic features (pitch, amplitude, spectral entropy) were also binarized. For different analyses in this paper, 0/1 can denote the behavioral feature that is below/above or above/below its median, or as not belonging/belonging to a specific percentile interval. The inset shows the area RA within the song pathway, two synapses away from the vocal muscles, from which these data were recorded.

are sufficiently large to be used as examples of dictionary reconstruction, allowing us to illustrate the type of biological insight that our approach can gain: we have 49 data sets — spanning 4 birds, 30 neurons and sometimes multiple song syllables for each neuron — for which we observed at least 200 instances of the behavior and the associated neural activity, which we estimate below to be the lower threshold for a sufficient statistical power.

To represent analysis of this motor code as a dictionary reconstruction problem, we binarize the recorded spiking time series so that $\sigma_t = (0, 1)$ indicates the absence or presence of a spike in a time slice of $[(t - 1)\Delta t, t\Delta t]$, see *Figure 2*. Thus each time interval is represented by a binary variable, and interactions among these patterns are described by over-represented or under-represented sequences of zeros and ones in the data. Using a complementary information-theoretic analysis, *Tang et al., 2014* showed that the mutual information between the neural spike train and various features of song acoustics peaks at $\Delta t = 1 \ldots 2$ ms. Thus, studying precise timing pattern codes means that we focus on $\Delta t = 2$ ms (our datasets are not large enough to explore smaller $\Delta t$) as discussed previously in *Tang*

*et al., 2014*. Detection of statistically significant codewords at this temporal resolution would both re-confirm that this neural motor code is timing based, consistent with previous analyses (*Tang et al., 2014*), as well as for the first time reveal the specific patterns that most strongly predict behavior. We focus on neural time series of length $T = 40$ ms duration preceding a certain acoustic syllable, which includes the approximate premotor delay with which neurons and muscles influence behavior (*Tang et al., 2014*). Thus the index $t$ runs between 1 and $T/\Delta t = 20$.

Since we are interested in words that relate neural activity and behavior, we similarly binarize the motor output (*Tang et al., 2014*), denoting by 0 or 1 different binary characteristics of the behavior, such as pitch being either below or above its median value, or outside or inside a certain range (*Figure 2*). We treat the behavioral variable as the 0'th component of the neuro-behavioral activity, which then has $N = 21$ binary variables, $\vec{\sigma} = \{\sigma_i\}_0^{N-1}$. Building the neural-behavioral dictionary is then equivalent to detecting significantly over- or under-represented patterns in the probability distribution $P(\vec{\sigma})$. Focusing specifically on the statistically anomalous words that include the behavioral bit results in detection of *codewords*, for which the neural activity is correlated with (or is predictive of) the behavioral bit. Note that $2^N = 2^{21} \approx 2 \cdot 10^6$, which is much greater than $M \sim 100 \dots 1000$ observations of the activity that we can record, illustrating the complexity of the problem. In fact, similar to the famous birthday problem (one gets coinciding birthdays with a lot fewer people than the number of days in a year), one expects a substantial number of repeating samples of the activity of the full length $N$ — and hence the ability to detect statistically over- and under-represented binary words – only when $M \sim \sqrt{2^N}$, which is what limits the statistical power of any dictionary reconstruction method. Crucially, the approach presented here works by analyzing all patterns, not just patterns of the full length, allowing us to detect anomalous sub-words even in more severely undersampled regimes.

## The unsupervised BIA method (UBIA) for dictionary reconstruction

To reconstruct dictionaries in the neural motor code dataset and others with similar properties, we have developed the unsupervised Bayesian Ising Approximation (uBIA) method based on the Bayesian Ising Approximation for detection of significant features in linear Gaussian regression problems (*Fisher and Mehta, 2015*). Specifically, we extend BIA to detect significant interaction terms in probability distributions, rather than in linear regression models. For this, we write the probability distribution $P(\vec{\sigma})$ without any loss of generality as

$$\log P(\vec{\sigma}|\vec{\theta}) = -\log Z + \sum_{i=0}^{N} \theta_i \sigma_i + \sum_{j \geq i}^{N} \theta_{ij} \sigma_i \sigma_j +$$
$$\sum_{k \geq j \geq i}^{N} \theta_{ijk} \sigma_i \sigma_j \sigma_k + \cdots + \theta_{0\dots N} \sigma_0 \times \cdots \times \sigma_N = \quad (1)$$
$$-\log Z + \sum_{\mu} \theta_{\mu} \prod_{i \in V_{\mu}} \sigma_i,$$

where $Z$ is the normalization coefficient (*Amari, 2001*). We use the notation such that $V_{\mu}$ is a nonempty subset of indexes $i \in [0, N]$, and $\mu = [1, 2^{N+1} - 1]$ is the subset number. Then $\{\theta_{\mu}\} = \vec{\theta}$ are coefficients in the log-linear model in front of the corresponding product of binary σs. In other words, $V_{\mu}$ denotes a specific combination of the behavior and / or times when the neuron is active. If $\theta_{\mu}$ is nonzero for a term $\prod_{i \in V_{\mu}} \sigma_i$, where $i = 0$ (the response variable) is in $V_{\mu}$, then this specific spike word is correlated with the motor output, and is a significant codeword in the neural code, see *Figure 1*. Finding nonzero $\theta_{\mu}$s is then equivalent to identifying *which* codewords matter and should be included in the dictionary in *Figure 1*, and inferring the exact values of $\theta_{\mu}$ tells *how* they matter. Notice that *Equation 1* makes precise the definition of the order of an interaction, which corresponds to the number of variables $\sigma_i$ in the interaction term.

A common alternative model of probability distributions uses $x = 2\sigma - 1 = \pm 1$ instead of $\sigma = (0, 1)$. A third order term coupling, for example, $\sigma_i \sigma_j \sigma_k$ represents a combination of first, second, and third order terms in the corresponding xs, and vice versa. Thus which words are codewords may depend on the parameterization used, but the longest codewords and nonoverlapping groups of codewords remain the same in both parameterizations. Our choice of σ vs $x$ is for a practical reason: a codeword in the σ basis does not contribute to $P$ unless *all* of its constituent bins are nonzero. Thus since spikes are rare, we do not need to consider contributions of very long words to the code.

We would like to investigate the neural dictionary systematically and without arbitrarily truncating *Equation 1* at some order of interactions or making other strong assumptions about the structure of the words in the dictionary. In fact, this is possibly the biggest distinction of our approach from others

in the literature (**Bialek et al., 1991**; **Pillow et al., 2008**; **Schneidman et al., 2006**), which usually start with strong a priori assumptions. However, as discussed above, some assumptions *must* be made to solve the problem for typical data set sizes, and we would like to be very explicit about those we make. To achieve all of this, we define indicator variables $s_\mu = (0, 1)$, $\mu = 1, \ldots, 2^{N+1} - 1$, which denote if a particular sequence of $\sigma_i = 1$, $i \in V_\mu$, and $\sigma_i = 0$, $i \notin V_\mu$, 'matters' (is a putative word in the dictionary), that is, it is either statistically significantly over- or under-represented in the data set compared to a null model (which we define later). In other words, we rewrite $P(\vec{\sigma}|\vec{\theta})$ without any assumptions as:

$$\log P(\vec{\sigma}|\vec{\theta}) = -\log Z + \sum_\mu \theta_\mu s_\mu \prod_{i \in V_\mu} \sigma_i. \tag{2}$$

We then choose a prior on $\theta_\mu$ and on $s_\mu$. We choose to focus on problems where there are many weak words in the dictionary; in other words, typically $|\theta_\mu| \ll 1$. We make this choice for two reasons. First, detecting words that are strongly anomalously represented is easy, and does not require a complex statistical apparatus. Second, having many contributing small effects is more realistic biologically. Specifically, for songbird vocal motor control, since many neurons control the muscles and hence the behavioral output, individual spikes in single neuron can only have a very weak effect on the motor behavior. We thus work in the *strong regularization limit* and impose priors

$$\mathcal{P}(\theta_\mu|s_\mu = 1) \propto \exp\left[-\tfrac{1}{2\epsilon}(\theta_\mu - \theta_\mu^*)^2\right], \ \epsilon \ll 1. \tag{3}$$

Note that the prior distribution $\mathcal{P}(\theta_\mu|s_\mu = 0)$ is irrelevant since, for $s_\mu = 0$, $\theta_\mu$ does not contribute to $P(\vec{\sigma}|\vec{\theta}, \vec{s})$.

At this point, we need to choose the null model for the occurrence of letter patterns. We do this by choosing $\bar{\theta}_\mu$ in a way such that the a priori averages (calculated within the prior only) and the empirical averages (frequencies, observed in the data) of individual $\sigma_i$s are equal, $\langle \sigma_i \rangle = \overline{\sigma_i}$ (we always use $\langle \ldots \rangle$ and ⎯ to denote a priori and a posteriori expectations, respectively). This is equivalent to saying that the null model reproduces the firing rate of neurons and the frequency of the behavior, $P_{\text{null}} = \prod_{i=1}^N P(\sigma_i)$. This is possible to do since, in typical problems, marginal probabilities $P(\sigma_i)$ are, indeed, well-known, and it is the higher order interaction terms, the words in the dictionary, that make the reconstruction hard. Finally, we choose the least informative prior $P(s_\mu = 1) = P(s_\mu = 0) = 0.5$, so that a priori a particular word has a fifty-fifty chance of being included in the neural dictionary. If we have reasons to suspect that some words are more or less likely to be included a priori, this probability can be changed.

Since we are only interested in whether a word is anomalous enough to be in the dictionary, but not in the full model of the joint probability distribution, we integrate out all $\theta_\mu$, after having observed $M$ samples of the $N$ dimensional vector $\vec{\sigma}$. To perform this calculation, we start with the Bayes formula (notice that for the whole set of $M$ samples of the vector $\vec{\sigma}$ we use the notation $\vec{\boldsymbol{\sigma}}$)

$$P(\vec{s}|\vec{\boldsymbol{\sigma}}) \propto P(\vec{\boldsymbol{\sigma}}|\vec{s}) = \int d\boldsymbol{\theta} \, P(\vec{\boldsymbol{\sigma}}|\vec{\theta}, \vec{s}) \prod_{\mu|s_\mu=1} \mathcal{P}(\theta_\mu|s_\mu = 1). \tag{4}$$

Now we make two approximations. First, we evaluate the integral in *Equation 4* using the saddle point approximation around the peak of the *prior*, $\vec{\theta}^*$. This is a low signal-to-noise limit, and it is different from most high signal-to-noise approaches that analyze the saddle around the peak of the *posterior*. This leads to

$$\log P(\vec{\boldsymbol{\sigma}}|\vec{s}, \epsilon) \propto -\tfrac{1}{2}\log|\mathbf{I} - \epsilon\mathbf{H}| + \tfrac{\epsilon}{2}\mathbf{b}^\mathsf{T}(\mathbf{I} - \epsilon\mathbf{H})^{-1}\mathbf{b}, \tag{5}$$

where $\mathbf{H}$ and $\mathbf{b}$ have size $S \times S$ and $S$ respectively, being $S = \sum_\mu s_\mu$ being the total number of active variables. Their elements corresponds to

$$b_\mu = \left.\frac{\partial \mathcal{L}}{\partial \theta_\mu}\right|_{\vec{\theta}^*},$$

$$H_{\mu\nu} = \left.\frac{\partial^2 \mathcal{L}}{\partial \theta_\mu \partial \theta_\nu}\right|_{\vec{\theta}^*}, \tag{6}$$

where $\mathcal{L} = \log P(\vec{\sigma}|\vec{\theta})$ is the log-likelihood. Second, we do all calculations as a Taylor series in the small parameter $\epsilon$ (see below on the choice of $\epsilon$). Both approximations are facets of the same strong regularization assumption, which insists that most coupling constants $\theta_\mu$ are small. Again, the logic here is that we may not have enough information to know what $\theta$ is a posteriori, but we should have enough to know if it is nonzero. Following *Fisher and Mehta, 2015*, we obtain

$$\log P(\vec{\sigma}|\vec{s}, \epsilon) \propto \tfrac{\epsilon}{2}\left(\text{Tr}[\mathbf{H}] + \mathbf{b}^\mathsf{T}\mathbf{b}\right) + \tfrac{\epsilon^2}{2}\left(\tfrac{1}{2}\text{Tr}[\mathbf{H}^2] + \mathbf{b}^\mathsf{T}\mathbf{H}\mathbf{b}\right). \tag{7}$$

Finally, after explicitly reintroducing We now explicitly reintroduce the indicator variables and by taking into account the both $\mathbf{H}$ and $\mathbf{b}$ are restricted to the dimensions where $s_\mu = 1$. That is, for example, the term $\mathbf{b}^\mathsf{T}\mathbf{b}$ corresponds to $\sum_\mu b_\mu^2 s_\mu$. Finally, adding the normalization, we get

$$P(\vec{s}|\vec{\sigma}, \epsilon) = \tfrac{1}{\mathcal{Z}(\epsilon)} \exp\left[\epsilon \sum_\mu h_\mu(\vec{\sigma})s_\mu + \epsilon^2 \sum_{\mu\nu} J_{\mu\nu}(\vec{\sigma})s_\mu s_\nu\right], \tag{8}$$

where the magnetic fields (biases) $h_\mu$ and the exchange interactions $J_{\mu\nu}$ are

$$h_\mu(\vec{\sigma}) = \left.\frac{1}{2}\left[\frac{\partial^2\mathcal{L}}{\partial\theta_\mu^2} + \left(\frac{\partial\mathcal{L}}{\partial\theta_\mu}\right)^2\right]\right|_{\vec{\theta}^*},$$

$$J_{\mu\nu}(\vec{\sigma}) = \left.\frac{1}{4}\left[\frac{\partial^2\mathcal{L}}{\partial\theta_\mu\partial\theta_\nu}\left(\frac{\partial^2\mathcal{L}}{\partial\theta_\mu\partial\theta_\nu} + 2\frac{\partial\mathcal{L}}{\partial\theta_\mu}\frac{\partial\mathcal{L}}{\partial\theta_\nu}\right)\right]\right|_{\vec{\theta}^*}, \tag{9}$$

see 'Geometric interpretation of uBIA Field' in 'Materials and Methods' for a geometric interpretation of the field.

Notice that *Equation 8* has a familiar pairwise Ising form (*Thompson, 2015*), with data-dependent magnetic fields $h_\mu$ and the couplings $J_{\mu\nu}$. This Ising model has $2^N$ spins, replacing the Ising model with $N$ spins, but with higher order interactions in *Equation 1*. Naively, we created a harder problem, with many more variables! However, since most of the $2^N$ words do not appear in the actual data, and because of the $\epsilon^2$ in front of the pairwise coupling term, evaluating posterior expectations $\langle s_\mu\rangle$ for all word that actually occur is relatively easy, as we show now. Indeed, plugging in the model of the probability distribution, *Equation 1*, we get for the fields and the exchange interactions

$$h_\mu = \frac{M^2}{2}\left[(\overline{\sigma}_\mu - \langle\sigma_\mu\rangle)^2 - \frac{\text{var}(\sigma_\mu)}{M}\right], \tag{10}$$

$$J_{\mu\nu} = \frac{M^2}{4}\text{cov}(\sigma_\mu, \sigma_\nu)\left[\text{cov}(\sigma_\mu, \sigma_\nu) - 2M(\overline{\sigma}_\mu - \langle\sigma_\mu\rangle)(\overline{\sigma}_\nu - \langle\sigma_\nu\rangle)\right]. \tag{11}$$

Here, to simplify the notation, we defined $\sigma_\mu \equiv \prod_{i\in V_\mu}\sigma_i$, and we remind the reader that $M$ represents the number of samples. Further, angular brackets, cov, and var denote the a priori expectations, covariances, and variances of frequencies of words in the null model, which matches frequency of occurrence of each individual $\sigma_i$ (probability of firing in every time bin for the songbird data). Similarly, overlines denote the empirical counts or correlations between co-occurrences of words in the observed data. Specifically, denoting by $n_\mu$ the marginal frequencies of the word $V_\mu$ in the data, these expectations and frequencies are defined as follows:

$$\overline{\sigma}_\mu = \frac{1}{M}\sum_{m=1}^{M}\left(\prod_{i\in V_\mu}\sigma_i^{\{m\}}\right) = \frac{n_\mu}{M}, \tag{12}$$

$$\langle\sigma_\mu\rangle = \prod_{i\in V_\mu}\langle\sigma_i\rangle = \prod_{i\in V_\mu}\frac{n_i}{M}, \tag{13}$$

$$\text{var}(\sigma_\mu) = \langle\sigma_\mu^2\rangle - \langle\sigma_\mu\rangle^2 = \langle\sigma_\mu\rangle\left(1 - \langle\sigma_\mu\rangle\right) = \prod_{i\in V_\mu}\frac{n_i}{M}\left(1 - \prod_{i\in V_\mu}\frac{n_i}{M}\right), \tag{14}$$

$$\text{cov}(\boldsymbol{\sigma}_\mu, \boldsymbol{\sigma}_\nu) = \langle \boldsymbol{\sigma}_\mu \boldsymbol{\sigma}_\nu \rangle - \langle \boldsymbol{\sigma}_\mu \rangle \langle \boldsymbol{\sigma}_\nu \rangle = \prod_{k \in V_\mu \cup V_\nu} \frac{n_k}{M} - \left( \prod_{i_\mu} \frac{n_i}{M} \right) \left( \prod_{j \in V_\nu} \frac{n_j}{M} \right), \tag{15}$$

To derive these equations, note that $\sigma_i^2 = \sigma_i$. Note also that $\text{cov}(\boldsymbol{\sigma}_\mu, \boldsymbol{\sigma}_\nu) = 0$ if the intersection of $V_\mu$ and $V_\nu$ is empty.

*Equation 10* has a straightforward interpretation. Specifically, if the difference between the a priori expected frequency and the empirical frequency of a word is statistically significantly nonzero (compared to the a priori standard error), then the corresponding word is anomalously represented. It does not matter whether the word is over- or under-represented: in either case, if the frequency deviates from the expectation, then the field $h_\mu$ is positive, biasing the indicator $s_\mu$ toward 1, and hence toward inclusion of the word in the dictionary. If the frequency is as expected, then the field is negative, and the indicator is biased towards 0, excluding the word from the dictionary. Note that as $M$ increases, the standard error goes down, and the field generally increases, allowing us to consider more words. The sign of $\theta_\mu$ would reflect whether the word is over- or underrepresented. However, estimating the exact value of $\theta_\mu$ from small datasets is often impossible and is not our goal, even though, in *Figure 2*, we denote words as under- or over-represented by whether their empirical frequency is smaller or larger than the a priori expectation. Thus in some aspects, our approach is similar to the previous work (*Schnitzer and Meister, 2003*), where multi-neuronal patterns are found by comparing empirical firing probabilities to expectations. However, we do this comprehensively for *all* patterns that occur in data. Crucially, in addition, the exchange interactions $J_{\mu\nu}$ also allow us to account for reducibility of the dictionaries.

To see this, recall that correlations among words create a problem since a word can occur too frequently not in its own right, but either (a) because its sub-words are common, or (b) it is a sub-word of a larger common word, as illustrated in *Figure 1*. In other approaches, resolving these overlaps requires imposing sparsity or other additional constraints. In contrast, the couplings $J_{\mu\nu}$ address this problem for uBIA naturally and computationally efficiently. Notice that because of the factor of 2 in the negative term in *Equation 11*, the exchange interactions are predominantly negative if one expects the two studied words to be correlated, and if they co-occur in the empirical data as much as they are expected to co-occur in the null model because of the overlaps in their composition, $V_\mu$ and $V_\nu$. Negative $J_{\mu\nu}s$ implement a mechanism, where statistical anomalies in data that can be explained, in principle, by many different $\theta_\mu s$ are attributed predominantly to one such $\theta_\mu$ that explains them the best, bringing the dictionary closer to the irreducible form. On the other hand, the exchange interactions are positive if one expects correlations between the words a priori, but does not observe them. Thus, in principle, a word can be included in the dictionary even at zero field $h_\mu$. Crucially, every word affects the probability of every other word to be included in the dictionary by means of their corresponding $J_{\mu\nu}$. In this way, while uBIA is not equivalent to the full maximum entropy definition of irreducibility (*Margolin et al., 2010*), it comes close.

Knowing the coefficients $h_\mu$ and $J_{\mu\nu}$, one can numerically estimate $\langle s_\mu \rangle$, the posterior expectation for including a word $V_\mu$ in the dictionary. Generally, finding such marginal expectations from the joint distribution in disordered systems is a hard problem. However, here $h_\mu \propto \epsilon$ and $J_{\mu\nu} \propto \epsilon^2$, so that the fields and the interactions create small perturbations around the 'total ignorance' solution, $\langle s_\mu \rangle = 1/2$ (this is a manifestation of our general assumption that none of the words is very easy to detect). Therefore, we calculate the marginal expectation using fast mean field techniques (*Opper and Saad, 2001*). We use the *naive* mean field approximation, which is given by self-consistent equations for the posterior expectations in terms of the magnetizations $m_\mu = 2\langle s_\mu \rangle - 1$,

$$\tanh^{-1}(m_\mu(\epsilon)) = \frac{\epsilon}{2} \left[ h_\mu + \epsilon \sum_\nu J_{\mu\nu} + \frac{\epsilon}{2} \sum_\nu J_{\mu\nu} m_\nu(\epsilon) \right] \tag{16}$$

$$= \frac{1}{2} \left[ \epsilon h_\mu + \epsilon^2 h_\mu^{\text{eff}}(\epsilon) \right], \tag{17}$$

so that interactions among spins are encapsulated in an effective field $\epsilon h_\mu^{\text{eff}}$. We solve *Equation 16* iteratively (*Fisher and Mehta, 2015*), by increasing $\epsilon$ from 0 —that is, from the total ignorance $\langle s_\mu \rangle = 1/2$ or $m_\mu = 0$ — and up to the limiting value $\epsilon_{\max}$ in steps of $\delta\epsilon = M^{-1}/20$. This limiting value $\epsilon_{\max}$ is

determined by the two approximations involved in the strong regularization assumption. First, the saddle point approximation around the peak of the prior in *Equation 4* implies that the characteristic width of the prior should be smaller than that of the likelihood, $\epsilon \leq \epsilon_1 = 1/M$. Second, the Taylor series up to second order in $\epsilon$ for the posterior of the indicator variables implies that the quadratic corrections should not be larger than the linear terms. Within the mean field approximation, this means that $\langle |h_\mu| \rangle_\mu \geq \langle |\epsilon h_\mu^{\mathrm{eff}}(\epsilon)| \rangle_\mu$, which is saturated at some $\epsilon_2$ (notice that, in contrast to our usual notation, the averages here are over the indices, and not the data). Thus, overall we take $\epsilon_{\max} = \min\{\epsilon_1, \epsilon_2\}$. In other words, we use the largest $\epsilon$ (the weakest possible regularization), which is still consistent with the strong regularization limit. Additionally we have used the TAP equations (*Opper and Saad, 2001*), instead of *Equation 16* to calculate magnetizations. These are more accurate since they account for how a spin affects itself through its couplings with the other spins. However, corrections due to this more complicated method were observed to be negligible in our strong regularized regime, since they were of higher order in $\epsilon \ll 1$. Thus, all results that we report here, and the software implementation of uBIA on GitHub, (copy archived at swh:1:rev:a374c0d478958aaf38415c7b616bbdebe83c6219) are based on the mean field estimation.

Note that the analysis above, and our GitHub code, only focused on words that appear in the data. However, most of the $2^N$ possible words must be absent from any realistic dataset. In the *Supplementary Materials*, we show that neglecting these words when calculating posterior probabilities for word inclusion does not lead to significant errors.

## Testing and fine-tuning uBIA on synthetic data

To verify that uBIA can, indeed, recover dictionaries and to set various adjustable parameters involved in the method, we tested the approach on synthetic data that are statistically similar to those that we expect to encounter in real-world applications, such as our neural recordings. We used the log-linear model, *Equation 1*, as a generative model for binary correlated observables $\vec{\sigma}$ with $N = 20$. While somewhat small compared to many state of the art experimental datasets, this choice of $N$ is highly relevant to the motor control studies, which are our primary target in this work. We chose the individual biases in the generative model from a normal distribution, $\theta_i \sim \mathcal{N}(-0.7, 0.1^2)$, which matched the observed probability of a spike in a bin in the bird data. That is, $p(\sigma_i = 1) \simeq [1 + \exp(-2\theta_i)]^{-1} \sim q \sim 0.2$. Then we selected which binary variables interacted. We allowed interactions of 2nd, 3rd, and 4th order, with an equal number of interactions per order. For different tests, we chose the interaction strengths from (a) the sum of two Gaussian distributions, one with a positive mean and the other with a negative one, $\mathrm{mean}(\theta_\mu) = \pm 0.5$, $\mathrm{std}(\theta_\mu) = 0.1$, and (b) from one Gaussian distribution centered at zero with $\mathrm{std}(\theta_\mu) = 0.5$. Both choices reflect our strong regularization assumption, so that effects of individual variables on each other are weak, and a state of one variable does not determine the state of the others, and hence does not 'freeze' the system. We were specifically interested in performance of the algorithm in the case where $\vec{\theta}$ are distributed as the sum of Gaussians. On the one hand, this tested how the algorithm deals with data that are atypical within its own assumptions. On the other hand, this choice ensured that there were fewer values of $\vec{\theta}$ that were statistically indistinguishable from zero, making it easier to quantify findings of the algorithm as either true or false. We have additionally tested other distributions of $\vec{\theta}$, but no new qualitatively different behaviors were observed. Finally, for both types of distributions of $\vec{\theta}$, we also varied the density of interactions $\alpha$ (number of interactions per spin), from $\alpha = 2$ to $\alpha = 4$, which spans the interaction densities of tree-like and 2D lattice-like networks. We generated $M$ samples from these random probability distribution and we applied our pipeline to reconstruct the dictionary. We tested on 400 distributions from each family. As the first step, we discarded high-order words absent in the data using a threshold on the expected number of occurrences $\langle \sigma_\mu \rangle M = \langle n_\mu \rangle < 0.02$. Next, we selected $N_{\max}$ words that have the highest (absolute) values in magnetic field $h_\mu$ (we have tested $N_{\max} = 200, 500, 2000, 5000$, and finally used 500 after not observing substantial differences). To decide which of these high-field words are to be included in the dictionary, we built the Ising model on the indicator variables, *Equation 8*, with its corresponding magnetizations $m_\mu$ given by the mean field equations. We started from an inverse regularization strength of $\epsilon = 0$ and then decreased the regularization strength by increasing $\epsilon$ in steps of $\delta\epsilon = 1/(20M)$, up to $\epsilon_{\max} = \min\{\epsilon_1, \epsilon_2\}$, as detailed above.

Next we needed to identify the significance threshold for the magnetization $m_\mu$, or, equivalently, for the posterior probability of including a word into the dictionary $\langle s_\mu \rangle = (m_\mu + 1)/2$. As is often the

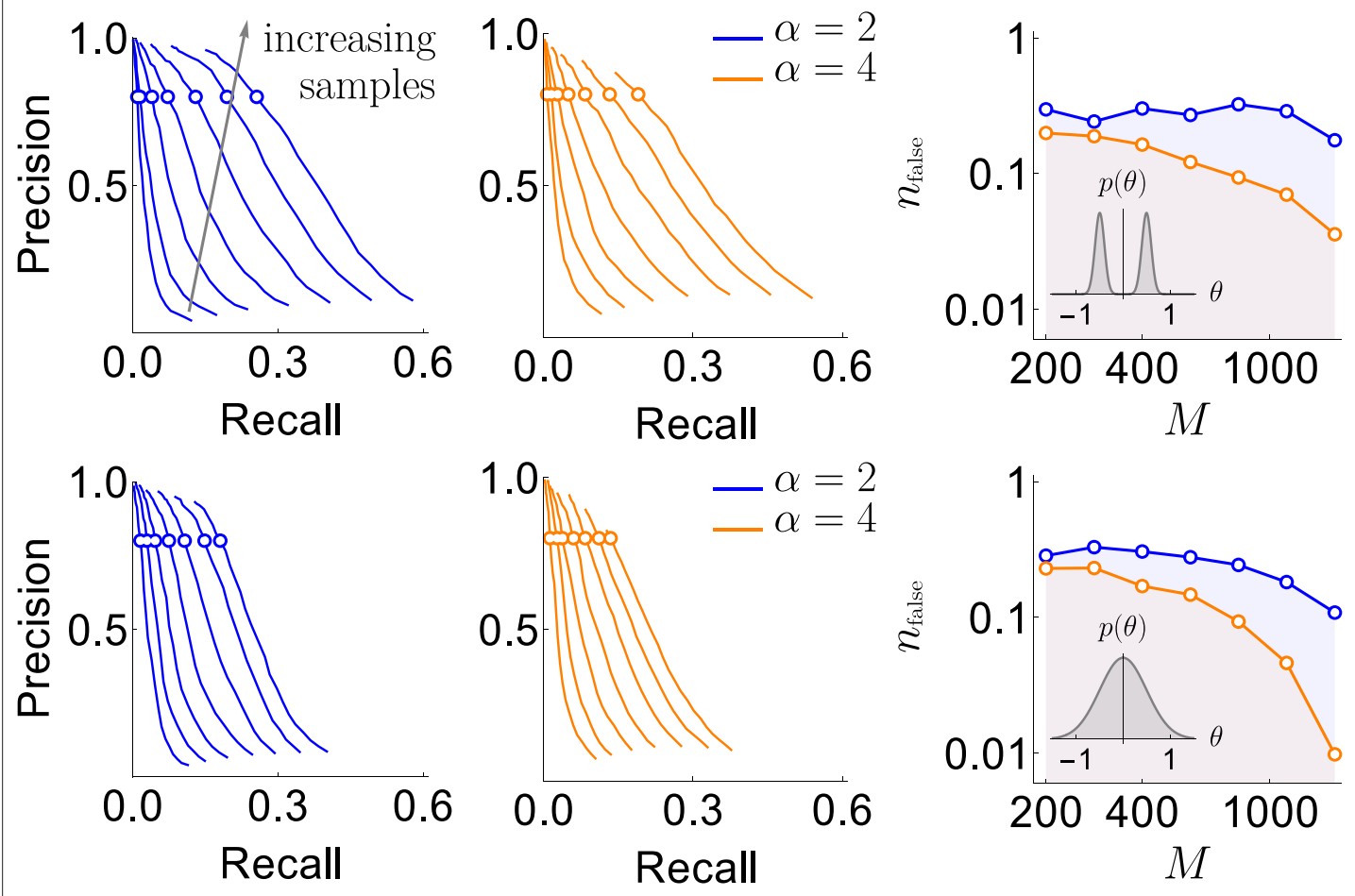

**Figure 3.** Results of the synthetic data analysis. Performance on synthetic data as a function of the density of interactions α, the distributions $p(\theta)$ for the strength of interactions, and the number of samples $M \in \{200, \ldots, 1600\}$ (logarithmically spaced). The first and the second columns correspond to precision-recall curves for the different density of interactions (significant words) per variable, $\alpha \in \{2, 4\}$, within the true generative model. The top and the bottom rows corresponds to the interaction strengths $\vec{\theta}$ selected from the sum of two Gaussian distributions, or a single Gaussian, as described in the text. For the first two columns, we vary the significance threshold in marginal magnetization $m(n_{\text{false}})$, such that the full false discovery rate on the shuffled data $n_{\text{false}} \in [0.005, 40]$. In the third column we show the value of $n_{\text{false}}$ that corresponds to the precision of 80% as a function of $M$ (the number of samples), so that the precision is larger than in the shaded region. This region is quite large and overlaps considerably for the four cases analyzed, illustrating robustness of the method.

case, this is affected by two contradictory criteria. Setting the threshold high would miss a lot of weakly significant words (high false negatives), but the words remaining in the dictionary would be likelier to be true (low false positives). In contrast, setting the threshold low would admit weakly significant words (lowering false negatives) at the cost of also admitting words by chance (increasing false positives). To measure false positives and negatives, we used two metrics: precision and recall. The precision, $\xi$, is the fraction of the words included in the dictionary that are true, that is, have a nonzero $\theta_\mu$ in the generated true model. The recall, $\eta$, is the fraction of the words in the generated model with $\theta_\mu \neq 0$ that were included in the dictionary. Fundamentally, there are no first principles arguments for the choice of the magnetization inclusion threshold, and thus we explored many different values of $m$ and infer the functions $\xi(m)$ and $\eta(m)$ for every data set explored. In **Figure 3**, we plot the precision vs. recall curves parametrically as a function of $m$. We see that, as the amount of data increases, the recall generally increases, though it remains small. However, since data set sizes are relatively small, we do not expect to detect all words, especially in the case where $\theta_\mu$ are allowed to be close to 0 in the generative model (Gaussian distributed). Thus we emphasize precision over recall in setting parameters of the algorithm: we are willing to not include words in a dictionary, but those words that we include should have a high probability of being true words in the underlying

model. It is thus encouraging that, in all tested cases, there was an underlying magnetization threshold that allowed for a high (e.g. 80%) precision to ensure that almost all of the words that we detected can be trusted to be true. Crucially, we see that the precision-recall curves are remarkably stable with the changing density of interactions. As a final point for interpreting these figures, we point out that $\eta$ is smaller when interactions coefficients are taken from a Gaussian centered at zero. However, one could argue that missing words with very small $\theta_\mu$ should not be considered a mistake: they are not significant words in the studied dictionary.

An additional way of measuring the accuracy of our approach is by exploring the full false discovery rate $n_{false}$ — the total number of dictionary words that are false positives, averaged over our usual 400 realizations of the training samples — produced by our algorithm on fully reshuffled data, where every identified word is false, by definition. We did this with reshuffling that kept the observed frequency of individual variables $n_i$ constant. We mapped out computationally the relation $n_{\text{false}}(m)$, which, together with $\xi(m)$ explored above, allowed to explore the dependence between $\xi$ and $m$. *Figure 3* shows that $\xi = 80\%$ corresponds to $n_{\text{false}} < 1$ for every data set that we have tried. Specifically, by keeping $n_{false}$ below 0.5 (only about half a word detected falsely, on average, in shuffled data), we can reach a precision as high as 80%, with the recall of 20% - 30% of the codewords depending on the number of samples, the distribution of $\vec{\theta}$, and $\alpha$. This shows that the findings of our approach are robust to variation of many parameters.

For the rest of this work, we set the magnetization threshold as a function of the false discovery rate, and we will admit words to the dictionary only when they have their marginal magnetizations $m_\mu > m(n_{\text{false}} = 0.5)$. With that, we are confident that our method produces dictionaries, in which a substantial fraction of words correspond to true words in the data, though the details of how many may depend on various (often unknown) properties of the data themselves, including with respect to patterns that are possible yet were never observed empirically.

To quantify the effect of interactions in among words in shaping the final dictionary, we check how many words with a field larger than the smallest field of a putative codeword corresponding to the kept codewords were discarded. Crucially, of such words with large magnetic field were We observe that $\sim 40\%$ of all such large-field words are removed from the final dictionary due to the word-word exchange interactions. This signifies that uBIA works as designed in identifying multiple words that can explain a statistical pattern and choosing the smallest subset of words able to explain it.

We finish this section with the following observation about the performance of uBIA in regimes that are even more severely undersampled than considered above, so that most of relatively long words only happen once or never in the data. In this regime, uBIA has two major strengths. First, uBIA analyzes putative codewords of arbitrary length, so that then it will detect short sub-patterns as codewords – and, in any reasonable dataset, at least some short sub-patterns will coincide – for example, there are only $N$ first order words formed by $N$ interacting units, and typically $M \gg N$. Second, uBIA detects not just over-representation of patterns, but also their under-representation. Thus, if a complex spike word happens once or does not happen at all, it may enter the uBIA dictionary as well precisely because it happens so infrequently.

## Reconstructing songbird neural motor codes

Having fine-tuned the method on synthetic data, we can now test it on a biological dataset. We applied uBIA to the songbird data, confirming the precise timing pattern nature of the code in this dataset. We then demonstrate the generality of the algorithm by applying it to different parameterizations of the data, which allows us to make surprising observations about control of exploratory vs typical renditions of the song by the birds. Notice that, for all applications below, we have to binarize the behavior (pitch). This inevitably results in the loss of resolution and the corresponding loss of codewords. However, such binarization is meaningful in the context of songbirds (*Tang et al., 2014*), and, crucially, it cannot lead to emergence of keywords where they would not exist otherwise.

### Statistical properties of the motor codes

We start with *Figure 4*, which explores the occurence of just two specific codewords found by uBIA that encode high-pitch renditions of syllables. Note that these codewords are, indeed, overrepresented together with the high pitch vocalizations. Analyzing if a particular word is correlated with an acoustic feature is, of course, not hard. However, detecting words that should be tested, without a multiple

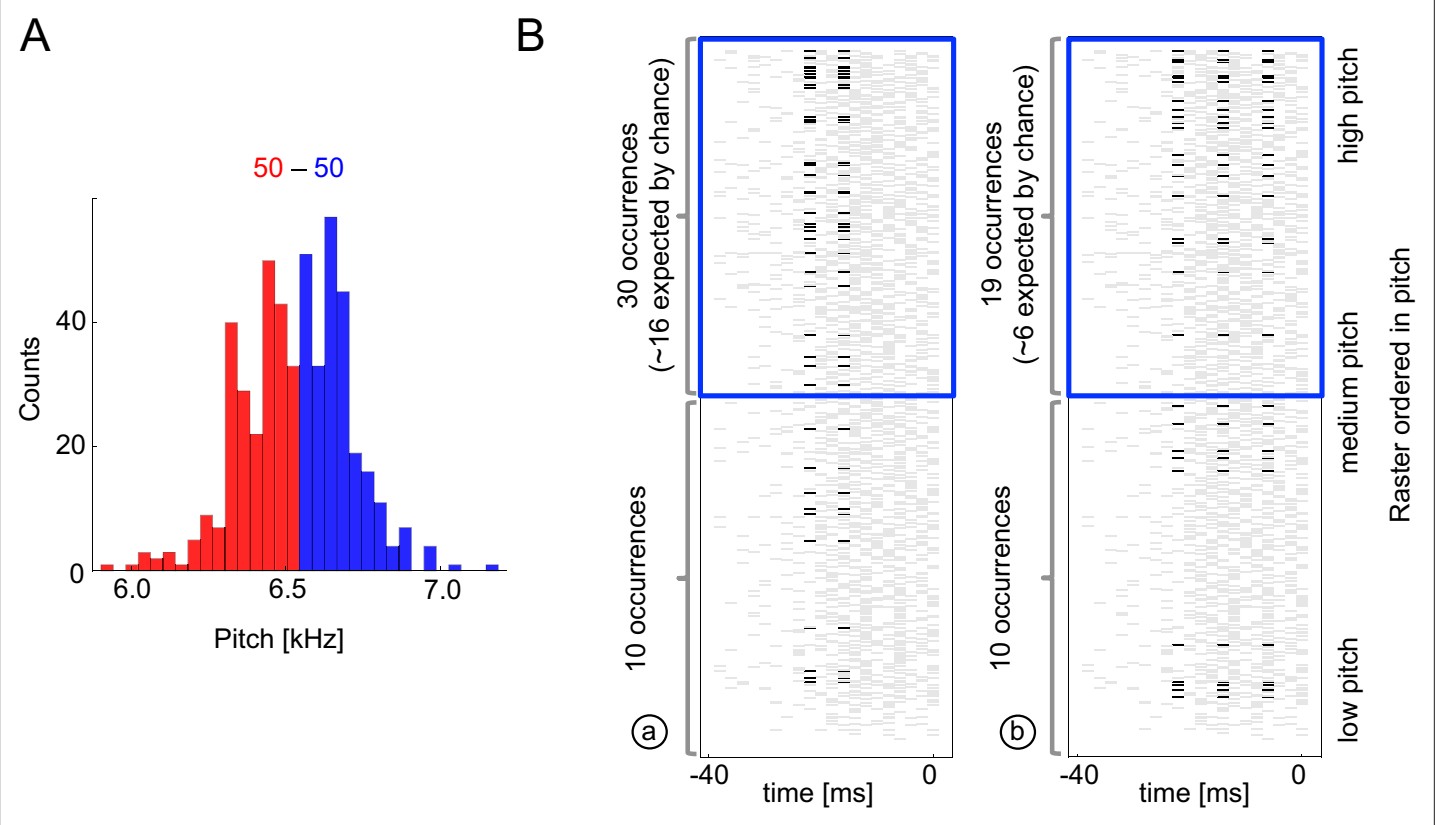

**Figure 4.** Sample multispike codewords. (**A**) Probability distribution an acoustic parameter (fundamental frequency, or pitch). For this analysis, we consider the output to be $\sigma_0 = 1$ when the pitch is above median (blue), and zero otherwise (red). (**B**) Distribution of two sample codewords (a two-spike word in the left raster, and a three-spike word in the right raster) conditional on pitch. In each raster plot, a row represents 40ms of the spiking activity preceding the syllable, with a grey tick denoting a spike. Every time a particular pattern is observed, its ticks are plotted in black. Note that these two spike words are codewords since they are overrepresented for above-median pitch (blue box) compared both to the null model based on the marginal expectation of individual spikes, and to the presence of the patterns in the low pitch region. Labels (a) and (b) identify these patterns in *Figure 5B*.

hypothesis testing significance penalty is nontrivial. Thus the power of uBIA comes from being able to systematically analyze abundances of *combinatorially many* such spike patterns, and further to identify which of them are *irreducibly* over- or under-represented. *Figure 5* illustrates statistical properties of entire neural-behavior dictionaries discovered by uBIA for different songbird premotor neurons and for three features of the acoustic behavior. While we reconstruct the dictionaries that include all irreducible words, including those that have only anomalous firing patterns but a statistically normal behavioral bit, here we primarily focus on codewords, which, recall, are defined as statistically anomalous relations between the behavior and the neural activity. We do the analysis twice, first for behavior binarized as $\sigma_0 = 1$ for the above-median acoustic features, and then for the below-median acoustic features. This way we detect words that predict either a behavior or its opposite. We do this because the same pattern of spikes should not be anomalous in the same way simultaneously when studying both the above and the below median codes, since the pattern cannot code for two mutually exclusive features. Detecting such patterns thus serves as a consistency check on our method. There were 0.7 such codewords on average per dictionary. This is consistent with the expected false discovery rate of $lt_1$ codewords per neuron for data sets of our size and statistical properties, further increasing our confidence in the method.

The most salient observation is that the inferred codewords consist of present or absent spikes in specific 2ms time bins, (*Figure 5A*). This is consistent with our previous analysis (*Tang et al., 2014*), which identified the same timescale for this dataset by analyzing the dependence of the mutual information between the activity and the behavior on the temporal resolution, but was unable to detect the specific words that carry the information. The second crucial observation is that most of codewords are composed of multiple spikes, (*Figure 5C*) representing an orthographically complex

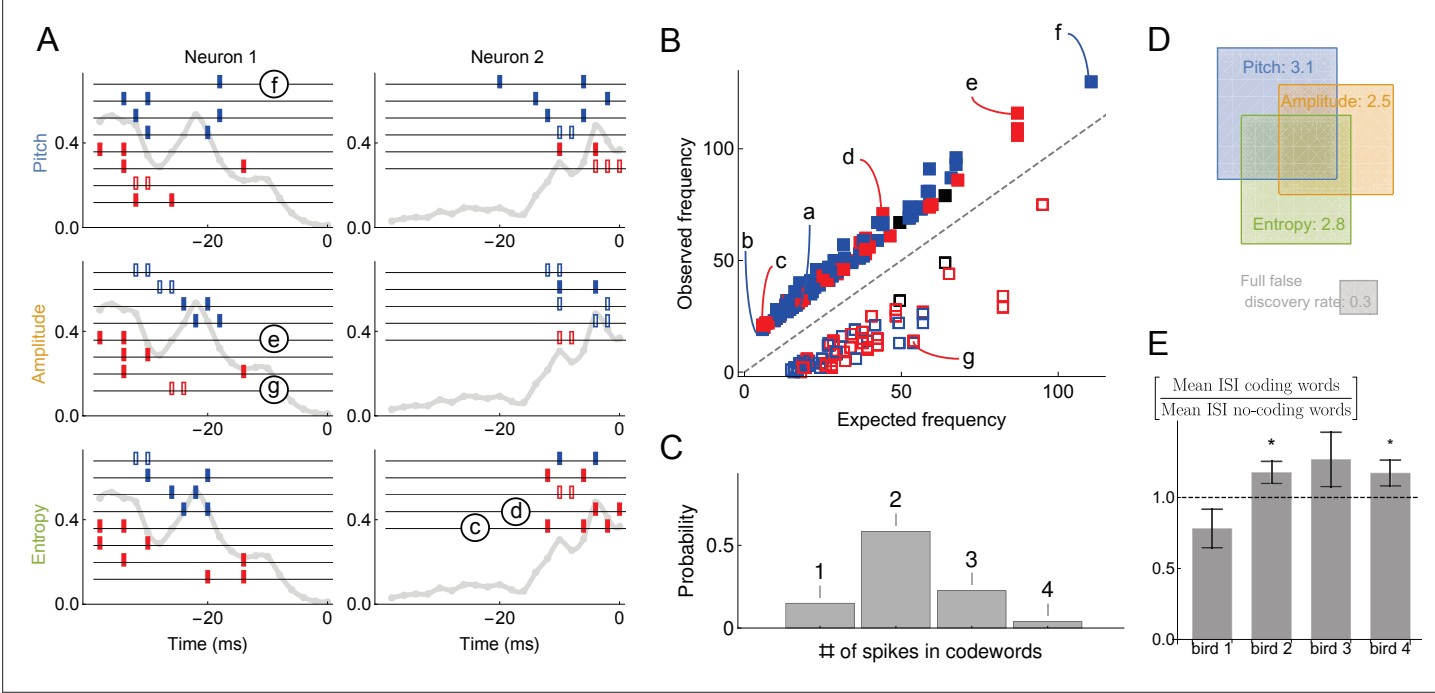

**Figure 5.** Statistical properties of neural dictionaries. (**A**) Sample neural-behavioral dictionaries for two neurons from two different birds (columns) and for three different acoustic features of the song (rows: pitch, amplitude, and the spectral entropy). The light gray curve in the background and the vertical axis corresponds to the probability of neural firing in each 2ms bin (the firing rate). The rectangular tics represents the timing of spikes in neural words that predict the acoustic features. For example, a two spike word with tics at points $t = i, j$ corresponds to the probability that the word $\mu = (i, j)$ is a codeword for the acoustic feature with a probability statistically significantly higher than 1⁄2. Codewords for high (low) output, that is, $\sigma_0 = 1$ above (below) the median, are shown in blue (red). Full (empty) symbols correspond to over(under)-occurrence of the codeword-behavior combinations compared to the null model. Finally full (empty) black symbols represent words that over(under)-occur in the blue code and under(over)-occur in the red code. Words labeled (c)-(g) are also shown in (**B**). (**B**) Frequency of occurrence of statistically significant codewords for different acoustic features in different neurons. Only first 200 codewords shown for clarity. Plotting conventions same as in (**A**), and letters label the same codewords as in (**A**) and in **Figure 4B**. (**C**) Proportion of $m$-spike codewords found in the dictionaries analyzed. An $m$-spike word corresponds to an $(m + 1)$-dimensional word in the neural-behavioral dictionary. Most of the significant codewords have two or more spikes in them. (**D**) Mean number of significant codewords, averaged across all neurons and acoustic features. An average neuron has 5.6 codewords in our dataset, of which 3.1 code for the pitch, 2.5 for the amplitude, and 2.8 for the spectral entropy, with the number of words coding for pairs of features or for all three of them indicated by the overlap of rectangles in the Venn diagram. For comparison, our estimated false discovery rate is 0.3 words, so that only ~0.3 spurious words are expected to be discovered in each individual dictionary. We note that about a third of all analyzed dictionaries are empty, so that those that have words in them typically have more than illustrated here. (**E**) Mean inter-spike interval (ISI) for the codewords (spike words that code for behavior) vs. all spike words that are significantly over- or under-represented, but do not code for behavior. Averages in each of the four analyzed birds are shown, illustrating that the ISI statistics of the coding and non-coding words are different, but the differences themselves vary across the birds. Star denotes 95% confidence. Other properties of the dictionaries (mean number of spikes in codewords, fraction of codewords shared by three vocal features, proportion of under/over-occurring codewords), do not differ statistically significantly across the birds.

*pattern* timing code (**Sober et al., 2018**), in contrast to single spike timing codes, such as in **Bialek et al., 1991**. Large number of codewords of 2 or more spikes (and thus 3 or more features, including the behavior itself) suggests that analyzing these dictionaries with the widely used lower order MaxEnt or GLM methods that typically focus on lower-order statistics (see *Online Methods*) would miss their significant components. Our third crucial observation is that very few sub-words / super-words pairs occur in the dictionaries (**Figure 5A**; e.g. the second codeword coding for entropy in neuron 2 in the panel A is a subword of the others). Finally, similarly to our synthetic data analysis, 30% of words with large magnetic field were removed from the final dictionary due to word-word interactions. This indicates that uBIA fulfills its goal of rejecting multiple correlated explanations for the same data.

We quantify these observations as follows. In the 49 different datasets, the average size of a dictionary within one dataset is 14 words. Of these words, on average 5.6 include the behavioral feature and hence are *codewords* (**Figure 5D**). That there are so many specific temporally precise codewords suggests that the behaviorally-relevant spike timing patterns are the rule, rather than the

exception, in this dataset. We found that 66% of codewords are unique to one of the three analyzed acoustic features. This further quantifies the observation that some neurons in RA are *selective* for specific acoustic features, as noted previously (*Sober et al., 2008*). Across all neurons and all acoustic features, only 15% of codewords consist of a single spike (or absence of spike), while 58%, 23%, and 4% consist of two, three, and four spikes respectively, (*Figure 5C*) (we are likely missing many long codewords, especially with small θ's due to undersampling). This observation is consistent across all neurons and acoustic features, again indicating that coding by temporally precise spike patterns is a rule and not an exception.

At the same time, the observed dictionaries are quite variable across neurons and the production of particular song syllables. Codewords are built by stitching together multiple spikes or spike absences, and individual spikes occur at certain time points in the (-40,0) ms window with different probabilities in different neurons and syllables (i.e. the firing rate is both time and neuron dependent, *Figure 5A*, grey lines). Codewords are likely to occur where the probability of seeing a spike in a bin is ~50%, since these are the times that have more capacity to transmit information. Thus variability in firing rates as a function of time across neurons necessarily creates variability in the dictionaries across these neurons. Beyond this, we observe additional variability among the dictionaries that is *not* explained by the rate fluctuations. For example, we can differentiate one of the four birds from two of the others just by looking at the proportions of high-order codewords (an average of 0.21 bits in Jensen-Shannon divergence between the target bird and the rest, which means that we need around five independent samples/codewords to distinguish this bird from the others). This is further corroborated by the fact that the mean inter-spike interval (ISI) for codewords is different from that of other words in the dictionaries, and this difference is also bird-dependent, see *Figure 5E*.

## Verification validation of the inferred dictionaries

To show that the dictionaries we decoded are biologically (and not just statistically) significant, we verify whether Statistical significance is not a substitute for biological significance. The only way to interpret findings of any statistical method, including ours, is through perturbation experiments. For example, one could try to see if a stimulation of a neuron with a specific codeword-like patterns would *cause* (rather than merely correlate with) a certain behavior (*Srivastava et al., 2017*). Unable to do this, we do a weaker validation and check if the codewords can, in fact, be used to predict the behavioral features. For this, we built two logistic regression models that relate the neural activity to behavior. The first one uses the presence / absence of spikes in individual time bins and the second the presence / absence of the uBIA detected codewords as predictor variables (see *Online Methods*). Note that the individual spikes model is still a precise-timing model, which has 20 predictors (20 time bins, each 2 ms long), and hence one may expect it to predict better than the codewords model, which typically has many fewer predictors. To account for the possibility of overfitting, in all comparisons we test the predictive power of models using cross-validation. We emphasize that we do not expect either of the two models to capture an especially large fraction of the behavioral variation. Indeed, *Tang et al., 2014* have shown that, at 2ms resolution, on average, there is only about 0.12 bits of information between the activity of an individual neuron and the behavioral features, and the assumption behind our entire approach is that none of individual predictors have strong effects. Further, a specific model, such as logistic regression, will likely recover even less predictive power from the data. With this, Figure 9 compares prediction between the two models, obtaining a significantly higher accuracy and a lower mean cross-entropy between the model and the data for the models that use codewords as predictors. In other words, the higher order, but smaller, dictionaries decoded by uBIA outperform larger, non-specific dictionaries in predicting behavior. This is especially significant since uBIA codewords are detected for their statistical anomaly and irreducibility, and not directly for how accurately they predict behavior.

## Dictionaries for exploratory vs. typical behaviors

Bengalese finches retain the ability to learn through their lifetimes, updating their vocalizations based on sensorimotor feedback (*Kuebrich and Sober, 2015*; *Kelly and Sober, 2014*; *Sober and Brainard, 2009*; *Saravanan et al., 2019*). A key element of this lifelong learning capacity is the precise regulation of vocal variability, which songbirds use to explore the space of possible motor outputs, (*Figure 6A and B*). For example, male songbirds minimize variability when singing to females during

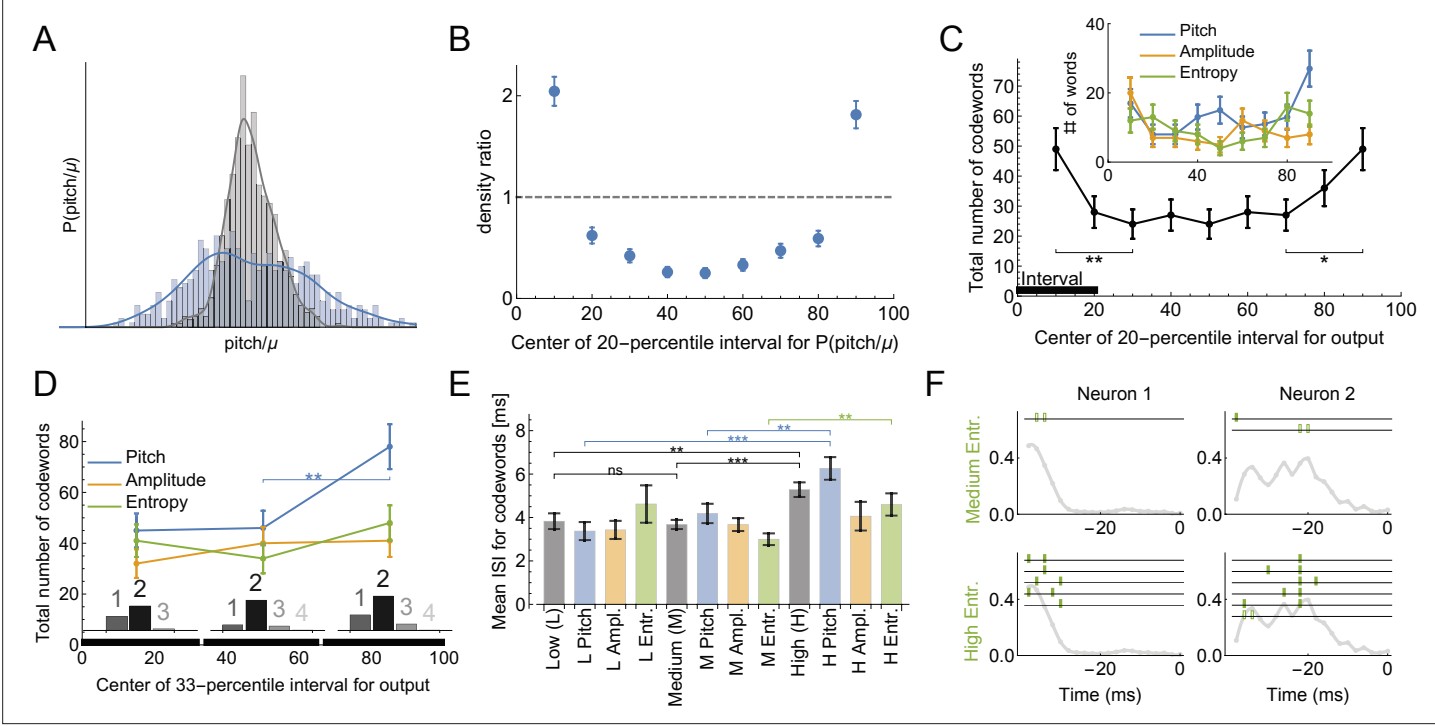

**Figure 6.** Codes for vocal motor exploration. (**A**) Distribution of syllable pitch relative to the mean for exploratory and performance behaviors (blue, intact birds, vs. grey, LMAN-lesioned animals, see main text). (**B**) Ratio of the histograms in (**A**) evaluated in the quintiles of the exploratory (blue) distribution centered around [10%,20,...90%] points. (**C**) Total number of codewords when considering the vocal output as 1 if it belongs to a specific 20-percentile interval of the output distribution, and 0 otherwise. We observe that there are significantly more codewords for the exploratory behavior (tails of the distribution compared to the middle intervals). Notice that the shape of the curves parallels that in (**B**), suggesting that exploration drives the diversity of the codewords. (**D**) Number of codewords when considering the vocal output as 1 if it belongs to a 33-percentile (non-overlapping) interval of the output distribution, and 0 otherwise. Here there are significantly more codewords when coding for high pitch. Further, the codewords found for each of the three intervals are mostly multi-spike (histograms show the distribution of the order of the codewords for each percentile interval). (**E**) For codewords for the 33-percentile intervals, we compare the mean inter-spike intervals (ISIs). Codewords for high outputs (especially for pitch and spectral entropy) have a significantly larger mean ISI. (**F**) We illustrate dictionaries of two neurons for the medium and the high spectral entropy ranges. Notice that the high entropy range has significantly more codewords.

courtship, but dramatically increase the range of variability in acoustic features such as pitch when singing alone (*Hessler and Doupe, 1999*; *Woolley et al., 2014*). The variability is controlled by the activity of nucleus LMAN. Silencing or lesioning LMAN reduces the acoustic variance of undirected song (*Figure 6A*) to a level approximately equal to that of female-directed song (*Kao et al., 2005*; *Olveczky et al., 2005*). Using uBIA, we can ask for the first time whether the statistics of codewords controlling the exploratory vs. the baseline range of motor variability are different. To do this, we analyze the statistics of codewords representing different parts of the pitch distribution. First, we define the output as $\sigma_0 = 1$ if the behavior belongs to a specific 20-percentile interval ([0 -20], [10 - 30], ..., [80 -100] ) and compare the dictionaries that code for behavior in each of the intervals. We find that there are significantly more codewords for exploratory behaviors (percentile intervals farthest from the median, *Figure 6C*). This holds true for different features of the vocal output, although the results are only statistically significant if pooled over all features. To improve statistical power by increasing the number of trials in each acoustic interval, we also consider a division of the output into three equal intervals: low, medium, and high. In this case, there are still more codewords for the high exploratory pitch, and the dictionaries for each of the intervals are still multispike (*Figure 6D*). We further observe that the codewords themselves are different for the three percentile groups: the mean ISI of high pitch, amplitude, and spectral entropy codewords is higher, with the largest effect coming from the pitch and the spectral entropy (*Figure 6E*). Examples of typical and exploratory dictionaries are illustrated in *Figure 6F*. Note that this analysis partially addresses the concern about losing resolution

due to discretization of behavior by exploring effects of different discretizations on the reconstructed dictionaries.

These findings challenge common accounts of motor variability, in songbirds and other systems, that motor exploration is induced by adding random spiking variations to a baseline motor program. Rather, the over-abundance of codewords in the exploratory flanks of the acoustic distributions indicates that the mapping between the neural activity and the behavior is more reliable than in the bulk of the behavioral activity: multiple renditions of the same neural command result in the same behaviors more frequently, making it easier to detect the codewords. One possibility is that the motor system is less biomechanically noisy for large behavioral deviations. This is unlikely due to the tremendous variation in the acoustic structure (pitch, etc.) of different song syllables within and across animals (*Sober and Brainard, 2009*; *Elemans et al., 2015*), which indicates that songbirds can produce a wide range of sounds and that particular pitches (i.e. those at at one syllable's exploratory tail) are not intrinsically different or harder for an animal to produce. Similarly, songbirds can dramatically modify syllable pitch in response to manipulations of auditory feedback (*Sober and Brainard, 2009*; *Kuebrich and Sober, 2015*). A more likely explanation for the greater prevalence of codewords in the exploratory tails is that the nervous system drives motor exploration by selectively introducing particular patterns into motor commands that are specifically chosen for their reliable neural-to-motor mapping. This would result in a more accurate deliberate exploration and evaluation of the sensory feedback signal, which, in turn, is likely to be useful during sensorimotor learning (*Zhou et al., 2018*).

Finally, although dissecting the role of different neural structures to generating code words would require additional (perturbation) experiments, we can speculate about the contributions of LMAN inputs and local RA circuitry to shaping the statistics of activity in RA. One possibility is that the greater prevalence of code words in exploratory behaviors reflects the interaction of unstructured variability (from LMAN) with the dynamics determined by local circuitry within RA. Alternatively, inputs patterns from LMAN during exploration might be highly structured across LMAN neurons such that tightly-regulated multi-spike patterns in LMAN, rather than the interaction of uncoordinated LMAN activity with intrinsic RA dynamics, is responsible for generating exploratory deviations in behavior. Future studies, including both perturbations of neural activity and recording ensembles of LMAN neurons, will shed light on these questions.

## Discussion

In this work, we developed the unsupervised Bayesian Ising Approximation as a new method for reconstructing biological dictionaries — the sets of anomalously represented joint activities of multiple components of biological systems. Inferring these dictionaries directly from data is a key problem in many fields of modern data-rich biological and complex systems research including systems biology, immunology, collective animal behavior, and population genetics. Our approach addresses crucial shortcomings that so far have limited the applicability of other methods. First, uBIA does not limit the possible dictionaries, either by considering words of only limited length or of a pre-defined structure, instead performing a systematic analysis through all possible words that occur in the data sample. Second, it promotes construction of irreducible dictionaries, de-emphasizing related, co-occurring words. Finally, uBIA does not make assumptions about the linear structure of dependencies unlike various linear methods.

To illustrate capabilities of the method, we applied it first to simulated data sets that are similar to those we expect in experiments. The method was able to reconstruct the dictionaries with a very low false discovery rate for a wide range of parameters and statistical properties of the data (*Figure 3*), which made us hopeful that uBIA's findings will be similarly meaningful in real-life applications. In this analysis, we explored the range $M \sim 10^2 \ldots 10^3$, and $N \sim 20$, which is highly relevant to the neurobiological data we focus on here. Crucially, this $N$ is smaller than in many modern high-throughput experiments. Indeed, there is a necessary trade-off among the system size, the amount of data, and the ability to explore the interactions systematically, to all orders. Since there are many methods able to analyze data at much larger $N$, but with making assumptions about, in particular, pairwise structure of words in the dictionary (see *Overview of prior related methods in the literature* in *Online Methods*), we decided to focus uBIA on systematic exploration of somewhat smaller systems, $N \sim 20$.

To show that the methoduBIA, indeed, can work with real data, we applied it to analysis of motor activity in cortical area RA in a songbird. We were able to infer statistically significant codewords from

large-dimensional probability distributions ($2^{21} = 2,097,152$ possible different words) with relatively small data sets ($\sim 10^2 \ldots 10^3$ samples). We verified that the codewords are meaningful, in the sense that they predict behavioral features better than alternative approaches. Importantly, most of words in hundreds of dictionaries that we reconstructed were more complex than is usually considered, involving multiple spikes in precisely timed patterns. The multi-spike, precisely timed nature of the codes was universal across individuals, neurons, and acoustic features, while details of the codes (e.g. specific codewords and their number) showed tremendous variability.

Further, we identified codewords that correlate with three different acoustic features of the behavior (pitch, amplitude, and spectral entropy), and different percentile ranges for each of these acoustic features. Across many of these analyses, various statistics of codewords predicting exploratory vs. typical behaviors were different. Specifically, the exploratory dictionaries contained more codewords than the dictionaries for typical behavior, suggesting that the exploratory spiking patterns are more consistently able to evoke particular behaviors. This is surprising since the exploratory behavior is usually viewed as being noisier than the typical one. Crucially, exploration is a fundamental aspect of sensorimotor learning (*Tumer and Brainard, 2007*; *Kuebrich and Sober, 2015*; *Kelly and Sober, 2014*), and it has been argued that large deviations in behaviors are crucial to explaining the observed learning phenomenology (*Zhou et al., 2018*). However, the neural basis for controlling exploration vs. typical performance is not well understood. Intriguingly, vocal motor exploration in songbirds is driven by the output of a cortical-basal ganglia-thalamo-cortical circuit, and lesions of the output nucleus of this circuit (area LMAN) abolishes the exploratory (larger) pitch deviations (*Kao et al., 2005*; *Olveczky et al., 2005*). Our findings therefore suggest that the careful selection of the spike patterns most consistently able to drive behavior may be a key function of basal ganglia circuits.

While the identified codewords are statistically significant, and, for the songbird data, we show that they can predict the behavior better than larger, but non-specific features of the neural activity, a crucial future test of our findings will be in establishing *biological* significance of uBIA findings. In the context of the neural motor control, biological significance may be in establishing the *causal* rather than merely correlative nature of the codewords, which can be done by stimulating neurons with patterns of pulses mimicking the codewords (*Srivastava et al., 2017*). Such verification will be facilitated by the speed of our method, which can reconstruct dictionaries in real time on a laptop computer. The speed and the relatively modest data requirements by uBIA will also allow us to explore how population-level dictionaries are built from the activity of individual neurons, how control of complex behaviors differs from control of their constituent features, how the dictionaries develop and are modified in development, and whether the structure of dictionaries as a whole can be predicted from various biomechanical and information-theoretic optimization principles.

In building uBIA, we have made a lot of simplifying assumptions. For example, in applications to neural populations, uBIA would explore only symmetric statistical correlations, while the physical neural connectivity is certainly asymmetric. Similarly, in the analysis of activity of a neuron over time bins in the current work, we did not account for causality. Further, we assumed that all data are stationary for the duration of the experiment, which will break down for longer experiments. Making useful biological interpretation of uBIA findings and designing better perturbation experiments may depend on our ability to lift some of these restrictions. The interpretation may be aided by extending uBIA to different null models. The current null model assumes independence between the units—a common assumption in the field. Detecting words that are represented anomalously compared to more complex null models, such as those preserving pairwise correlations, may focus the analysis on the most surprising, and hence maybe easier to interpret, codewords. This is a feasible future extension of uBIA. However, one would have to be careful, since, for datasets with $M \sim 10^2 \ldots 10^3$, pairwise correlations are not known well themselves, which may introduce additional uncertainty into the interpretation.

We tested uBIA on a small number of biological and synthetic datasets. However, for our method to be broadly applicable, users will need to adjust our the hyperparameters, for example, the significance threshold $m$ and the inverse regularization strength $\epsilon$, depending on the statistical structure of their data. Different values may be required depending on the dataset size and the prevalence and strength of higher-order interactions. Although it is hard to say a priori what these adjustments might be, we offer a few suggestions. First, we note that false negatives (failure to identify a significant

pattern) should not be considered a failure point major problem in the undersampled regime, since it is manifestly impossible to observe all patterns with relatively small datasetspossible coding patterns with little data. More worrying would be false positive errors, in which statistically insignificant patterns are identified as code wordscodewords. However, our algorithms algorithm offers a self-consistency check: a single pattern should not be identified as predicting both the presence and the absence of a behavior. One can search for such errors by relabeling the presence as 0 or 1 and repeating the analysis, see *Figure 5A*. Hyperparameters should therefore be adjusted to keep the number of such cases, and with them the rate of all false positives, below an acceptable threshold More generally, one can do similar tests by relabeling individuals variables 0→1 —words and their partial negations should not be both anomalously represented. The fraction of such incorrectly identified words is a good measure of the false positives rate. Then one should adjust the detection threshold $m$ to the smallest value that keeps the false discovery rate acceptable to the user.

Another important parameter is the inverse regularization strength $\epsilon$. We would like to keep it as large as possible, so that the regularization is the weakest. At the same time, our perturbative analysis depends crucially on $\epsilon$ being small. This suggests a trade-off for choosing $\epsilon$: make it as large as possible, but such that the perturbative constraints, discussed after *Equation 17* are satisfied. This value will also depend on a specific dataset and cannot be predicted a priori.

Finally, additional evidence of biological significance of the method will need to come from its application to other types of data, such as, in particular, molecular sequences, or species abundances in ecology. Crucial for this will be the match between the assumptions of our method (e.g. no dominant words) and the actual data in specific applications: there is no way to say a priori when this will happen, and one will simply need to try. Crucially, in all cases, if the method works, we expect it to be fast and to work well even for problems with large $N$. In part, this is because the accuracy of the method does not collapse for undersampled problems (large $N$ and not too large $M$, *Figure 3*), and its computational complexity is limited not by $N$, but by the number of distinct words that occur in data.

## Materials and methods
### Overview of prior related methods in the lterature

As we pointed out in the main text, a number of different methods have been developed for reconstructing various biological dictionaries, or for the related problem of building the model of the underlying probability distribution. It is important to compare and contrast uBIA to these methods in order to highlight when it should be used. Since these prior methods have been especially common in neuroscience, and since our main biological application throughout this article is also in neuroscience, this is where we will focus our comparisons.

For many different experimental systems, it has been possible to measure the information content of spike trains (*Fairhall et al., 2012*; *Tang et al., 2014*; *Srivastava et al., 2017*), but the question of decoding – which spike patterns carry this information and how? – has turned out to be a much harder one. Most of the effort has been expended on decoding per se: building the model of the activity distribution, rather than deciding which specific spike patterns should belong to the model. Multiple approaches have been used, whether in the context of sensory or motor systems, starting with linear decoding methods (*Bialek et al., 1991*). All have fallen a bit short, especially in the context of motor codes in natural settings, where an animal is free to perform any one of many behaviors it wishes, and hence statistics are usually poor, with only a few samples per behavior. A leading method is Generalized Linear Models (GLMs) (*Paninski, 2004*; *Pillow et al., 2008*; *Gerwinn et al., 2010*), which encode the rate of spike generation from a certain neuron at a certain time as a nonlinear function of a linear combination of past stimuli (sensory systems) or of future motor behavior (motor systems) and the past spiking activity of a neuron and its presynaptic partners. GLM approaches can detect the importance of the timing of individual spikes and sometimes interspike intervals for information encoding, but generalizations to detect importance of higher order spiking patters are not yet well established. Another common approach is based on maximum entropy (MaxEnt) models (*Schneidman et al., 2006*; *Granot-Atedgi et al., 2013*; *Savin and Tkačik, 2017*). These replace the true distribution of the data with the least constrained (i. e., maximum entropy) approximation consistent with low-order, well-sampled correlation functions of the distribution. The In some versions of the method, one then searches for the constraints that affect the distributions the most, and only focuses on those, thus

avoiding overfitting (*Barrat-Charlaix et al., 2021*). The MaxEnt approach is computationally intensive, especially when higher order correlations are constrained by data. As a result, almost all of the applications focus on, at most, constraining pairwise activities in the data. At the same time, to approximate empirical distributions well, a large number of such constraints is constraints—even just pairwise ones—is often required. This requires very large datasets, especially if one is interested in relating the neural activity to the external (behavioral or sensory) signals. Such large datasets are hard to obtain in the motor control setting. More recently, feed-forward and recurrent artificial neural network approaches have been used to decode large-scale neural activity (*Pandarinath et al., 2018*; *Glaser et al., 2017*), but these have focused primarily on neural firing rates over large (tens of milliseconds) temporal windows, and typically require larger datasets than considered here.

As a result of the large data set size requirement and of the focus on building the model of the neural activity rather than finding statistically anomalous features in it, to date, there have not been successful attempts to reconstruct neural dictionaries from data. A success method must (i) resolve spike timing in words of the dictionary to a high temporal resolution, (ii) be comprehensive and orthographically complex, not limiting the words to just single spikes or pairs of spikes, and (iii) discount correlations among spiking words to produce irreducible dictionaries that only detect those codewords that cannot be explained away by correlations with other words in the dictionary.

There are methods (*Ganmor et al., 2015*; *Prentice et al., 2016*) that are more closely related to our approach, and which can be used within the same pipeline as uBIA, potentially for even better results —specifically in an undersampled regime. When modeling high-dimensional data using too few samples, all such methods tend to decrease complexity of their fitted models (*MacKay, 1992*), such as using fewer parameters or sparser dictionaries. In this regime, such methods can be improved if there is some a priori information regarding which elements of the model must be fitted first. Here, uBIA could be invaluable: it can be used first to choose the features to be included in models, and then a complementary algorithm can be used to actually construct a model on these feature, not unlike we did with the logistic regression in this work. For example, *Ganmor et al., 2015* proposed to understand ganglion cell population activity in relation to the stimuli they encode, and they used a pairwise MaxEnt approach to model the (undersampled) probability distributions of neural activities conditional on specific stimuli. Alternatively, uBIA can be used first to identify conditional neural dictionaries (which will include only a subset of neural pairs, but also potentially higher order combinations of neurons). Then the conditional MaxEnt models can be build based on such detected conditional dictionaries, potentially alleviating the undersampling problem. Crucially, the same approach can applied to other models, not just the log-linear model, *Equation 1*, or the MaxEnt model. Indeed, as long as the log-likelihood function is specified within a model, and the derivatives of the log-likelihood with respect to the model parameters can be evaluated, similar to *Equation 9*, then we can use uBIA to calculate the fields and the exchange interactions, and eventually the probability of inclusion of any term in the model.

On the other hand, as the number of samples increases, we might want to increase the complexity of the involved models by adding additional explanatory terms. Choosing which terms to add is a combinatorially complex problem, and one generally wants to add terms that are non-redundant. Here, uBIA can be useful as well by providing a broad picture of how various terms in the more complex models interact, and hence biasing us towards growing the model complexity in complementary, rather than competing directions. For example *Prentice et al., 2016* considered a Hidden Markov model to describe ganglion cell activity, where different hidden modes activated at different time points. For a fixed mode, the probability distribution of the neural population activity was modeled as belonging to an exponential family. Before adding a new mode, one can calculate the uBIA exchange interactions between it and the already existing modes, as in *Equation 9*. If the exchange interactions are negative, the new mode is (partially) redundant with the existing one, and should probably be skipped in favor of another, more independent mode. Again, as long as the likelihood function can be written analytically and is differentiable with respect to the model parameters, uBIA can be applied in this way.

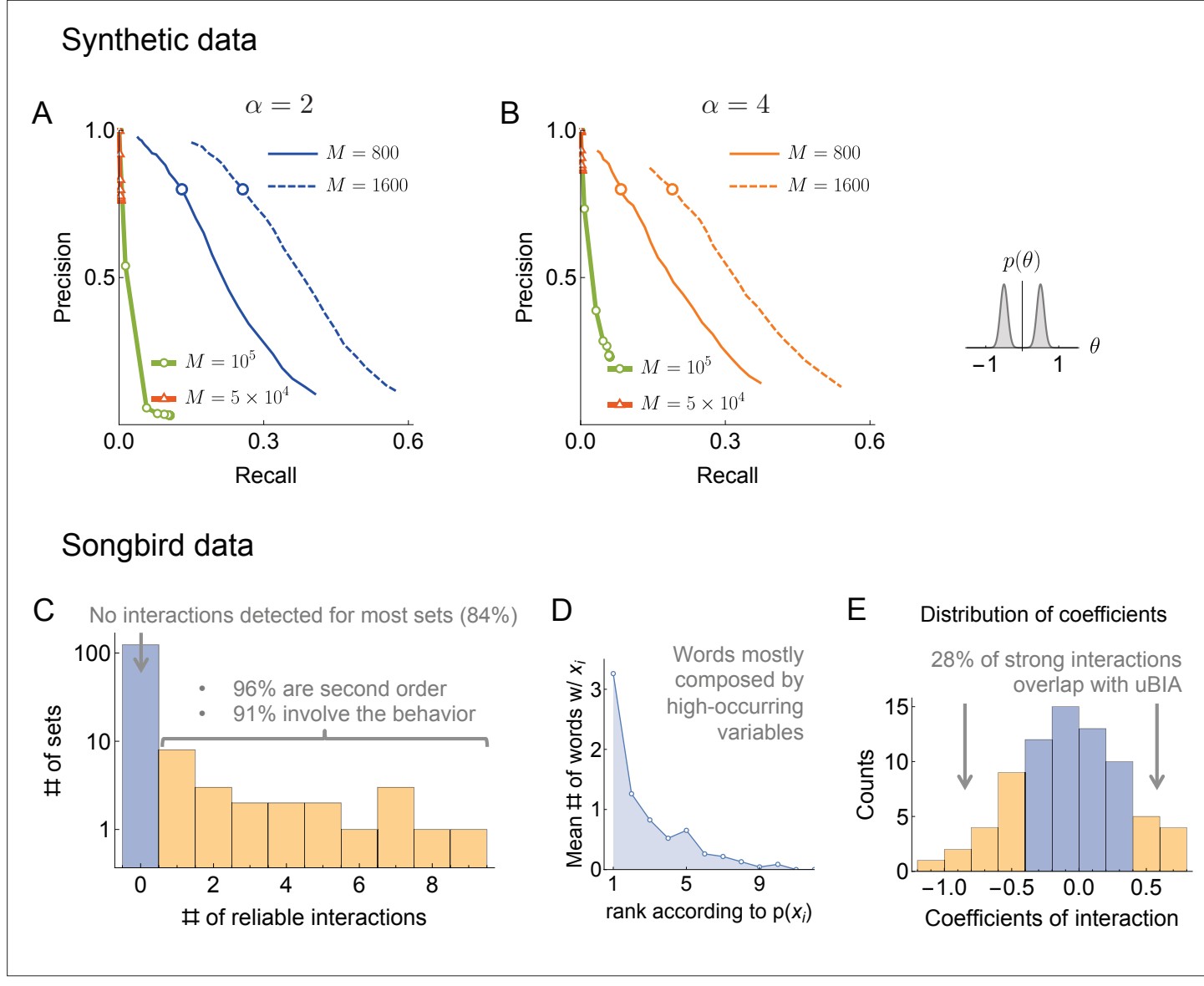

**Figure 7.** Comparison of Ganmor et al.method to uBIA. (**A, B**) For the synthetic data that we consider in the paper (*Figure 3*, interactions arising from the sum of two Gaussians), we obtained precision vs recall curves for the Ganmor et al. method (green and red) using a sweep over the absolute value of the inferred interaction threshold and comparing the detected interactions to the true ones. We also show the corresponding uBIA curves (blue) from *Figure 3* for $M = 800, 1600$. As illustrated, the Ganmor approach requires two orders of magnitude more data to begin discovering interactions and still does not reach the performance of uBIA for datasets with realistic sizes. (**C**) For the songbird data, the Ganmor et al. approach did not detect any interactions for most datasets. Of the 82 interactions that were detected, most corresponded to pairwise interaction between the behavior and the time bin. (**D**) Words identified by the Ganmor et al. were largely detected based on high marginal probability, consistent with an inability to detect higher order patterns directly. (**E**) The most significant detected interactions (largest interaction coefficients) generally overlap with words detected by uBIA.

## Direct application of MaxEnt methods to synthetic and experimental data

To illustrate that traditional MaxEnt methods for creating generative models do not work in our undersampled data regime, we apply the methods described in *Ganmor et al., 2015* to both our synthetic dataset and to our data collected from songbirds. First, note that the Ganmor method assumes that activity is dominated by a single (silent) state and then detects words in a hierarchical fashion. Specifically, higher order patterns (i.e. deviations from the silent state) cannot be detected unless all constituent lower order patterns have already been shown to be statistically significant. This

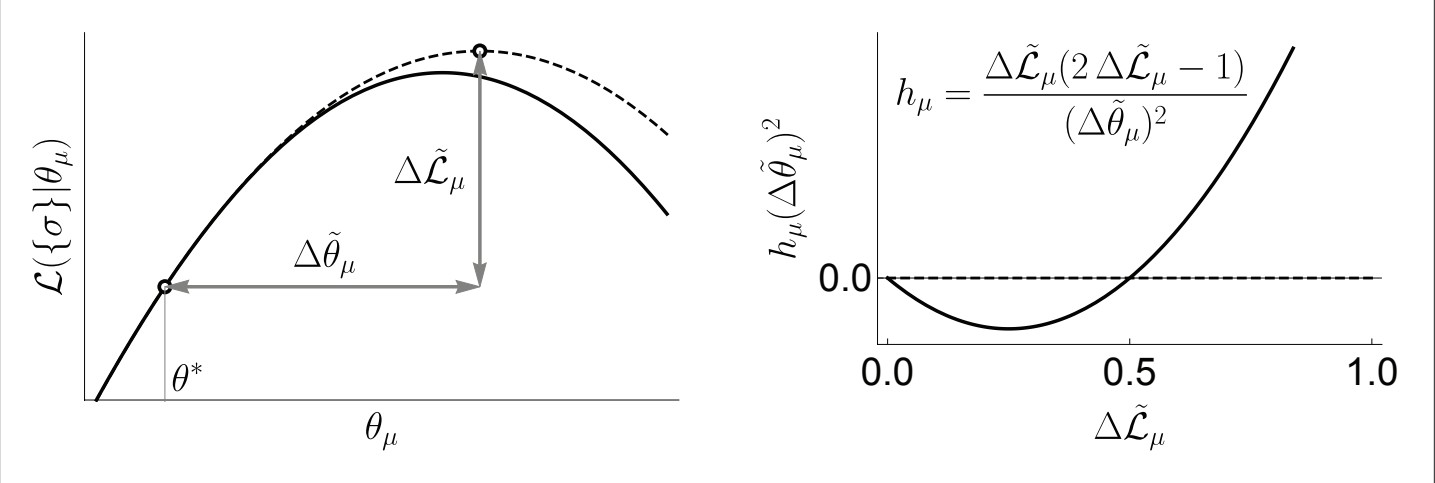

**Figure 8.** Geometric interpretation of the fields $h_\mu$ in the uBIA method, in relation to the log-likelihood function $\mathcal{L}(\{\vec{\sigma}\}|\vec{\theta})$. The uBIA method makes an approximate guess (dashed line in left panel) of how much in log-likelihood $\Delta\tilde{\mathcal{L}}_\mu$ we would win be fitting a parameter $\theta_\mu$, and how far in parameter space we would need to go, $\Delta\tilde{\theta}_\mu$ (see left panel). The sign of the field only depends on the improvement in log-likelihood, being positive beyond a threshold (inclusion of a word). This complexity penalty comes from the Bayesian approach in this strong regularized regime. On the other hand, the farther we go in parameter space, the smaller in absolute value the field becomes (see right panel).

requires datasets much larger than our approach, which can identify higher-order patterns directly, as illustrated in *Figure 7*.

## Geometric interpretation of uBIA field

The geometric interpretation of the $h_\mu$ fields in *Equation 9* is illustrated in *Figure 8*, and it showcases a part of how uBIA weights the addition of new parameters in terms of the improvement of the fitting, but without the need of building a explicit model. In relation to a null model located at $\theta^*$, the inclusion of a new parameter $\theta_\mu$ in general will improve the value of the log-likelihood $\mathcal{L}$. This improvement would have an approximated value of $\Delta\tilde{\mathcal{L}}_\mu$ and it would require to move from $\theta^*$ a distance $\Delta\tilde{\theta}_\mu$. The sign of the field $h_\mu$, which indicates the presence or absence of the parameter $\theta_\mu$, is determined by how big is the improvement, and the magnitude of the field by how far you need to move to fit such parameter. Then a rather small parameter that provides an acceptable improvement to the log-likelihood will have a high field.

## Effect of absent words

To calculate posterior expectations of inclusion of words, we focus only on words that appear in a specific dataset. There are many more words that do not, and the effect of these absent words on uBIA results must be analyzed.

We start with noticing that, of the exponentially many possible words, majority do not happen in a realistic data set. In particular, this includes most of long words. At the same time, a priori expectations for the frequency of such words, *Equation 13*, decrease exponentially fast with the word length. Thus the fields, *Equation 10*, for the words that do not occur are small, and the posterior expectation for including these words in the dictionary is $\langle s_\mu \rangle \approx 1/2$, so that we do not need to analyze them explicitly. A bit more complicated is the fact that all words affect each other's probability to be included in the dictionary through the exchange couplings $J_{\mu\nu}$, so that, in principle, the sum in the mean field equations, *Equation 17*, is over exponentially many terms. Thus it is possible for the absent words collectively to have a significant effect on the probability of inclusion of more common words into the dictionary. Here, we show that this collective effect on the interaction terms is exponentially small in $N$, as long as the empirical averages $n_i/M \ll 1$.

To illustrate this, we start with the probabilities $p(\sigma_i = 1) = p_i$ of a single variable being active. We then define the average such probability $q = N^{-1} \sum_i p_i$. Without the loss of generality, we assume $q < 1/2$, and otherwise we rename $\sigma_i \rightarrow 1 - \sigma_i$. Denoting a long word of a high order $k$ that does not occur in the data as $\boldsymbol{\sigma}_\omega$, we have $n_\omega = 0$. Then the corresponding field is

$$h_\omega = \frac{M^2}{2}\left[(0-\langle\boldsymbol{\sigma}_\omega\rangle)^2 - \frac{\text{var}(\boldsymbol{\sigma}_\omega)}{M}\right] \tag{18}$$

$$\sim -\frac{M}{4}q^k\left[1-q^kM\right] \sim -\frac{M}{4}q^k. \tag{19}$$

Here, we consider as *high order* words those, for which $q^kM \ll 1$ (in general, $\langle\sigma_\omega\rangle M = \langle n_\omega\rangle \ll 1$, which happens for $k \sim 4\ldots5$ for our datasets). Then the magnetization is

$$m_\omega(k) \simeq \tanh\left(\frac{\epsilon}{2}h_\omega\right) \sim -\frac{\epsilon M}{8}q^k. \tag{20}$$

This illustrates our first assertion that none of these non-occurring words will be included in the dictionary. However, as a group, they may still have an effect on words of lower orders. To estimate this effect, for a word $\boldsymbol{\sigma}_\mu$ of a low order $k_0$, we calculate the effective field $\tilde{h}_\mu^{\text{eff}}$, which all of the non-occurring words $\boldsymbol{\sigma}_\omega$ have on it. First we notice that, if $V_\mu$ and $V_\omega$ do not overlap, then their covariance is zero, and $J_{\mu\omega} = 0$. That is, only high-order words that overlap with $V_\mu$ can contribute to $\tilde{h}_\mu^{\text{eff}}$. Since $\text{cov}(\boldsymbol{\sigma}_\mu, \boldsymbol{\sigma}_\omega) \sim q^k(1-q^{k_0})$, the couplings are

$$J_{\mu\omega}(k) \sim \frac{M^2}{4}q^{2k}(1-q^{k_0}) \times \mathcal{O}(1). \tag{21}$$

Using **Equation 16**, this gives for the typical effective field that absent words have on the word μ

$$\tilde{h}_\mu^{\text{eff}}(\{\omega\} \to \mu) \simeq \epsilon \sum_{k \gtrsim k_0}^N \mathcal{N}(k)\,J_{\mu\omega}(k)\,(1+\frac{\epsilon}{2}m_\omega(k)), \tag{22}$$

where the number of words of order $k$ that overlap with $\boldsymbol{\sigma}_\mu$ and can affect it is given by the combinatorial coefficient $\mathcal{N}(k) \simeq \binom{N-k_0}{k-k_0}$. This has a very sharp peak at $k = (N+k_0)/2$, where $\mathcal{N} \simeq 2^{N-k_0}$. We can approximate the sum in **Equation 22** as the argument of the sum evaluated at this peak $k = (N+k_0)/2$, obtaining an effective field coming from high order words

$$\tilde{h}_\mu^{\text{eff}}(\{\omega\} \to \mu) \sim \epsilon\, 2^{N-k_0}\frac{M^2}{4}q^{N+k_0}(1-q^{k_0})\left[1-\frac{\epsilon^2 M}{16}q^{\frac{N+k_0}{2}}\right] \tag{23}$$

$$\sim \frac{\epsilon M^2}{4}\left(\frac{q}{1/2}\right)^N\frac{q^{k_0}}{2}(1-q^{k_0}) \tag{24}$$

$$\propto \left(\frac{q}{1/2}\right)^N. \tag{25}$$

In other words, even the combined effect of all higher order absent words is small if the average frequency of individual letters is smaller than 1/2. We thus can disregard all non-occurring words in the mean field equations.

We stress that, for this to hold, the average of the binary variables $\sigma_i$ must be small, $q = N^{-1}\sum_i p(\sigma_i = 1) < 1/2$. In our songbird dataset, this condition was fulfilled with $q \sim 0.2$. However, in 4% of cases the probability to have a spike in a certain time bin was $p_i > 1/2$. Thus to stay on the safe side, we performed additional analyses by redefining variables as $\sigma_i \to 1 - \sigma_i$ if the presence of a spike in a bin was >50 %. In other words, in such cases, we defined the absence of the spike as 1 and the presence as 0. For our datasets, the findings did not change with this redefinition.

This previous analysis does not imply that absent words of high order are irrelevant — it only says that they cannot be detected with the available datasets. In the numerical implementation of the method, we filter out long absent words $\omega$ such that $\langle\sigma_\omega\rangle M = \langle n_\omega\rangle < 0.02$, with this cutoff determined by **Equation 18-20**, so that, for these words, $h_\omega \ll 1$. These words get assigned 1/2 as the posterior probability of inclusion in the dictionary, and their contribution to the mean field equations is neglected. In contrast, if a word $\omega$ is absent but $\langle n_\omega\rangle \geq 0.02$, we include them in the analysis, **Equation 8**. Such words may turn out to be relevant code words, especially if they happen a lot less frequently than expected a priori.

### Testing the predictive power of the uBIA dictionaries

In this section, we test whether the codewords found in data from songbird premotor neurons can be used to predict the subsequent behavior. We compare two logistic regression models: one that uses the activity in the 20 time bins to predict the behavior and another that only uses as features the activity of the few relevant codewords, usually far fewer than 20. The features corresponding to the codewords are binary, and they are only active when all the time bins of such words are active. This means that the model using the time bins is more complex, as it already has all the information that the codewords model has and more, though it does not account for combinatorial effects of combining spikes into patterns. In order to properly test the predictive power between these two models with different complexity we perform twofold cross-validation, using a log-likelihood loss function. As is common in these cases, an L2 penalty is included to help the convergence to a solution (the models were implemented with the Classify function from Mathematica, whose optimization is done by the LBFGS algorithm). As shown by *Tang et al., 2014*, not all neurons in our dataset are timing neurons, or even code for the behavior at all. Thus we restrict the comparison to those cases that have at least 4 codewords (27 case in total, with 10 codewords on average). Both of the logistic regression models have the following structure

$$p(y = 1|z, \boldsymbol{\beta}) = \frac{1}{1 + \exp(-\beta_0 - \sum_i \beta_i z_i)}, \tag{26}$$

where $y$ corresponds to the behavior, and the features correspond to the time bins in one case ($z_i = x_i$) and to the codewords in the other ($z_i = \prod_{j \in V_i} x_j$), while $\beta_i$ are the coefficients of the model. The loss function used is the log-likelihood with the L2 penalty,

$$\mathcal{L} = \sum_{m=1}^{M} \log p(y^{(m)}|z^{(m)}, \boldsymbol{\beta}) - \frac{\lambda}{2} \sum_i \beta_i^2, \tag{27}$$

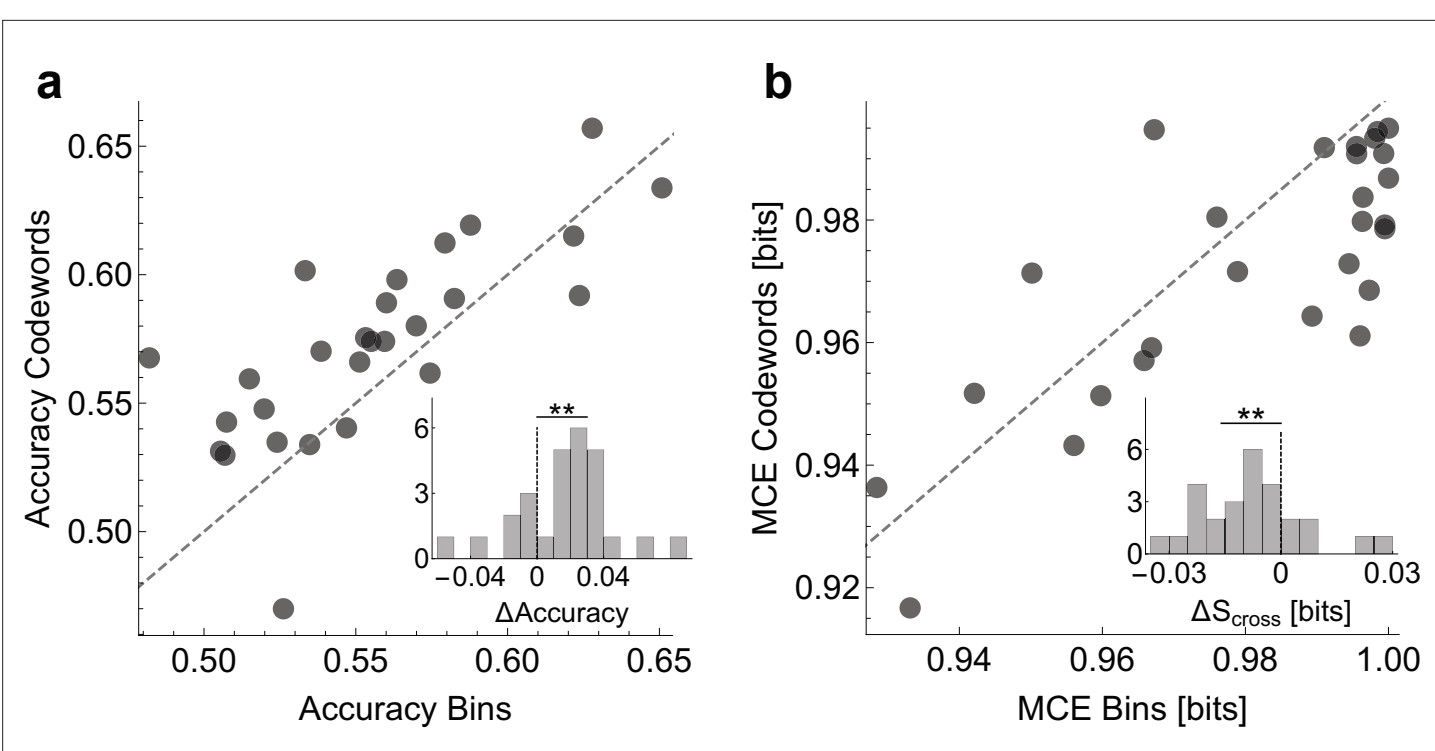

**Figure 9.** Prediction accuracy with uBIA dictionaries. We compare prediction of the behavior using logistic regression models that have as features (i) neural activity in all the time bins at 2ms resolution versus (ii) only the detected relevant codewords. (**A**) Scatter plot of accuracy of models of both types, evaluated using twofold cross-validation. Inset shows that the different between the prediction is significant with $p < 0.01$ according to the paired t-test. (**B**) Scatter plots of the mean cross-entropy between the data and the models for the two model classes. Inset: Even though the models that use the codewords are simpler (have fewer terms), they are able to predict better (with lower cross-entropy) according to the paired t-test.

where $M$ is the number of samples, and $\lambda$ is the regularization strength. In our analysis, as different datasets have different number of samples, we show the results for the mean cross-entropy over the test data, which correspond to the normalized log-likelihood.

*Tang et al., 2014* showed that individual neurons on average carry around 0.12 bits at a 2ms scale. So for both models, we expect the prediction accuracy to be barely above chance, especially since we are focusing on a particular prediction model (a logistic regression), and may be missing predictive features not easily incorporated in it. *Figure 9a* shows the scatter plot of accuracy in the 27 analyzed datasets, plotting the prediction using the time bins activity on the horizontal axis versus prediction using only the codewords activity on the vertical one. We observe that the models based on codewords are consistently better than the ones using all the 20 time bins, and the difference is significant (inset). We additionally evaluate the quality of prediction using the mean cross-entropy between the model and the data. *Figure 9b* shows that the models with the codewords have lower mean cross-entropies and thus generalize better (see Inset).

## Acknowledgements

We thank David Hoffman and Pankaj Mehta for valuable discussions. This work was supported in part by NIH Grants R01-EB022872, R01-NS084844, and R01-NS099375, NSF grant BCS-1822677 (CRCNS Program), and a grant from the Simons Foundation as part of the Simons-Emory International Consortium on Motor Control. IN acknowledges hospitality of the Kavli Institute for Theoretical Physics, supported in part by NSF Grant PHY-1748958, NIH Grant R25GM067110, and the Gordon and Betty Moore Foundation Grant 2919.01. IN and SJS further acknowledge hospitality of the Aspen Center for Physics, which is supported by NSF grant PHY-1607611.

## Additional information

### Funding

| Funder | Grant reference number | Author |
| --- | --- | --- |
| National Institutes of Health | R01-EB022872 | Damián G Hernández<br>Samuel J Sober<br>Ilya Nemenman |
| National Institutes of Health | R01-NS084844 | Samuel J Sober<br>Ilya Nemenman |
| National Institutes of Health | R01-NS099375 | Damián G Hernández<br>Samuel J Sober<br>Ilya Nemenman |
| National Science Foundation | BCS-1822677 (CRCNS Program) | Samuel J Sober<br>Ilya Nemenman |
| Simons Foundation | The Simons-Emory International Consortium on Motor Control | Samuel J Sober<br>Ilya Nemenman |
| Simons Foundation | Simons Investigator in MPS | Ilya Nemenman |

The funders had no role in study design, data collection and interpretation, or the decision to submit the work for publication.

### Author contributions

Damián G Hernández, Conceptualization, Formal analysis, Investigation, Software, Writing - original draft, Writing - review and editing; Samuel J Sober, Ilya Nemenman, Conceptualization, Formal analysis, Funding acquisition, Investigation, Writing - original draft, Writing - review and editing

### Author ORCIDs

Damián G Hernández http://orcid.org/0000-0002-8995-7495
Samuel J Sober http://orcid.org/0000-0002-1140-7469
Ilya Nemenman http://orcid.org/0000-0003-3024-4244

Decision letter and Author response
Decision letter https://doi.org/10.7554/eLife.68192.sa1
Author response https://doi.org/10.7554/eLife.68192.sa2

## Additional files

### Supplementary files
• Transparent reporting form

### Data availability

The software implementation of uBIA is available from GitHub (copy archived at swh:1:rev:a374c-0d478958aaf38415c7b616bbdebe83c6219). The data used in this work is available from figshare.

The following dataset was generated:

| Author(s) | Year | Dataset title | Dataset URL | Database and Identifier |
|---|---|---|---|---|
| Hernandez D | 2019 | Songbird premotor dictionaries | https://doi.org/10.6084/m9.figshare.10315844.v1 | figshare, 10.6084/m9.figshare.10315844.v1 |

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
