## [Editor Report]

This work introduces a new unsupervised Bayesian method for identifying important patterns in neural population responses. The method offers improvements relative methods that do not assume correlations in neural responses, and is likely to also outperform methods that take into account pairwise correlations in neural responses.

---

## [Decision Letter]

**Decision letter after peer review:**

[Editors’ note: the authors submitted for reconsideration following the decision after peer review. What follows is the decision letter after the first round of review.]

Thank you for submitting your work entitled "Unsupervised Bayesian Ising Approximation for revealing the neural dictionary in songbirds" for consideration by *eLife*. Your article has been reviewed by 2 peer reviewers, and the evaluation has been overseen by a Reviewing Editor and a Senior Editor. The reviewers have opted to remain anonymous.

Our decision has been reached after consultation between the reviewers and your reviewing editor. It took an unusually long time to reach this decision due to both personal and global health emergencies. We are sorry for the delay. Based on these discussions and the individual reviews below, we regret to inform you that this manuscript will not be considered further for publication in *eLife* as a Research Article.

However, your reviewing editor and reviewers have agreed that they would be willing to consider a revised version of the paper, reformatted as a "Tools and Resources" paper, if you choose to resubmit it as such to *eLife*. That revised paper should address fully the methodological and technical questions raised by the reviewers and your reviewing editor in this letter. As it stands, the reviewing team did not feel that they had enough information to properly judge the merits of the method, and that would be reassessed if a new and revised Tools and Resources manuscript is submitted.

The individual reviewer comments are appended below and can be used either as directives for that new submission, or as helpful pointers for a submission elsewhere. Summary thoughts are included here, with the detailed reviews from the 2 reviewers included at the end.

The paper details a new approach to learning the input/output relationships in complex biological systems that drive things like motor behavior or (perhaps) protein folding and other processes. The work focuses on birdsong as an example of a complex motor task potentially modulated by precise temporal patterns of spiking in the bird motor cortex (RA). The method claims to be able to use very few jointly recorded samples of the input and output to learn both a model for the distribution of the spiking patterns observed as well as how they drive behavior. The method uses strong priors in a Bayesian Ising Approximation to identify the neural code words that are expressed at unusually high or low frequency with respect to the behavioral output. The authors find a large number of spike patterns correlated with behavioral variations in syllable pitch. The method has potential applications beyond birdsong; the ability to fit input/output relationships with very few samples is appealing.

The reviewing team, however, had some major reservations about the results and felt that the current draft did not give them enough details to properly evaluate the method. A summary of those major points is given here:

1) The method:

The reviewers felt that there wasn't enough detail presented to clearly analyze the results of the model. Much of the detail was in the SI and Methods, where the exposition wasn't clear or detailed enough for the reader to properly assess the results.

Questions raised included:

– What's the null model?

– How does this method scale to larger neural populations?

– Is the clustering used here justified? Can it be compared to other, simpler clustering methods?

– How do you justify the choice of prior and how are the unknowns fit from the data?

– Other extant methods for finding reliable interactions in exponential models were not referenced or compared.

– Artificial data used in the SI are based on properties of the recorded neurons, and as the authors explain the cells seem to be quite weakly coupled, so the relevance of the model for other data is not clear.

2) The behavioral results:

The reviewers were not convinced by the pitch results, primarily because of the coarse-graining of pitch into just two categories. Again, much of the explanation was relegated to the Methods part of the paper and could have been presented more clearly. If this method works, this is clearly interesting, but the results presented seemed rather weak.

3) The format of the paper:

The paper read a bit like 3 manuscripts glued together, each of which has merits, but none of which seemed to satisfactorily stand on its own. The paper starts with a grand introduction that talks about solving a major challenge in quantitative biology. Then it quickly dives into something that reads more like a methods paper, but with insufficient exposition and clarity. Finally, the paper shifts to the application of the method to behavior and neural data, the results of which seemed somewhat weak and preliminary.

Taking these critiques together, the reviewing team felt that the paper could not be reasonably revised for *eLife* as a Research Article. However, the team did find the central method potentially interesting if more details are provided that allow for proper assessment of it. The team felt that it could be a good "Tools and Resources" advance, if the method holds up to this more detailed scrutiny.

*Reviewer 2:*

The paper is a potentially interesting application of a recent statistical method, the Bayesian Ising Approximation (Fisher and Mehta), aimed at addressing the problem of estimating a complex distribution, in particular the relationship between a code and an output, from very few samples. This is accomplished by heavily weighting by the prior. The authors have applied this method to support the idea that there are precisely timed spike patterns that are especially significant in the generation of behavior, in this case in the output of syllables of birdsong. They use the method to identify "code words", patterns that are claimed to appear with notably high or low frequency in correspondence with a behavioral variable. They apply this method to find a larger number of precise spike patterns correlated with behavioral variations in the tails of the behavioral distribution. They suggest an interpretation that perhaps larger exploratory motor gestures require a more precise spiking code; although it might also be argued that this shows that there are more ways to achieve these larger fluctuations? While the claim is counterintuitive, it rests on the power of the method, whose exposition left a number of gaps.

1. It is unclear to what extent this goal is meaningful in the case that the paper analyses: premotor drive of muscles in the birdsong pathway. It seems evident that timing of RA spikes necessarily sculpt motor output; and the authors have previously analyzed this in detail to reveal timing precision (Tang et al.). It was not clear, however, that specific patterns meaningfully emerge from a sea of similar spike trains to encode exact outputs. The authors ask whether precise patterns in 2ms bins have a significant relationship with binarized song characteristics in 40ms bins-a very low fidelity relationship on the "behavior" side compared with a high level of precision on the neural side. This seems a dramatic discrepancy between the dimensionality of neural response vs behavior that deserves more justification. How general a space of coding relationships (nonmonotonic, etc) does this method explore given this binarization?

2. Further, it was unclear whether a temporally structured firing rate would be equally convincing as a representation of behavioral variation. While the concept of "irreducible dictionaries" is appealing, does this method fairly compare a timing code to a rate code? For example, doesn't the competition component of the model suppress the identification of firing rate-like contributions? Looking at Figure 1, the cartoon of eliminating overlapping but similar code words seem equivalent to ignoring a less temporally precise method of coding for behavior. That is to say, if you are forcing your model to choose one of several similar code words in a winner-take all inclusion choice, isn't the timing precision you see partially a factor of not looking at the other similar code words which didn't win out? The comparison of this method to GLM methods of relating the spiking to behavior is unclear. One alternative model was analyzed in the online methods, but is not clearly explained and such comparisons deserve to be treated in the main text. It should also be easy to report the simple 0th order analysis of how well you can predict the binary version of behavior simply with a spike count within the 40 ms window, or with some intermediate level of temporal variation. Seeing these comparisons as a control of the assertion that precise timing matters in this system would enhance intuition for the necessity of this method and better justify the claim "that they predict behavioral features better than other approaches" (line 256).

3. In general, the exposition of the method leaves many questions and could be improved considerably in clarity. I have tried to summarize the logic of the paper's core argument below and underscored throughout where more help could be given to understand the meaning and generality of assumptions made.

Logic of the calculation:

The paper sets up the problem of writing down a model for the joint probability of a behavioral event s0 and the firing of a spike in each, i, of N bins, σi. The authors write this in terms of an expansion over every possible combination of σi,

Log P(σ|θμ) = -log Z + Σ θμ Π _{i in Vμ} σi.

Q3.1: please expand a little on the exponential form. In maximum entropy approaches, this form of distribution arises from the maximum entropy solution, but if we are avoiding MaxEnt approaches, can you briefly justify it? In this approach, a firing pattern with spikes in bins (1,2,3,4) is considered to be independent of the events (1,2) and (3,4); whereas in maxent, if one is only expanding up to order 2, one hypothesizes that the probability of the word (1,2,3,4) will be absorbed into the pairwise probabilities, and one can ask how different the data is from that guess. It seems that here one is parametrizing with vastly more parameters-one for every possible spike/behavior combination-- without a systematic method of cutting them down (or at least one that was clear to me; it is addressed at the end of the Online Methods calculations once we have moved into the s variables but it would be helpful to get some intuition from the start).

One then defines "indicator variables" sμ which tell us which patterns appear significantly less or more often than one would expect under some null model. It would be good to specify this null model right away, and in straightforward terms; I don't see it defined in the main text at all.

Q3.2: Please explain at this point the null model for the frequency of appearance of any possible pattern of spikes and its correlation with behavior? Should it not depend on the specific behavior?

It is unclear why one is now allowed to write that the original definition of the model is EQUAL to the model now written including only terms defined by a criterion defined by our subjective interest in them:

Log P(σ|θ) = -log Z + Σ θμ sμ Π _{i in Vμ} σi

Q3.3: Why does one not here need to define a new and different probability distribution, something that makes some claim about belonging to the codeword dictionary, in which these terms are the only ones that appear?

Now the authors turn to the task of estimating the 2N parameters θμ. Instead of doing this estimation directly from the data (i.e. the a posteriori expectation), they use a prior on these values: they choose to assume that each of the values are narrowly distributed about some mean θ∗μ:

P(θ∗μ : sμ=1) ~ exp(-(θμ −θ∗μ)2).

This distribution is taken only to hold if the word is indeed a code word, ie statistically associated with a behavior. If it is not a code word, the prior is not defined.

Q3.4: Please justify this choice in more detail. Why are the epsilons not dependent on μ?

Q3.5: The argument for where these 2N values of θ∗μ come from is very unclear. Please spell out more clearly how constraining the means of the individual σis provides sufficient constraints for all 2N values, or why this is not needed.

Q3.6: Why does a code word have an a priori likelihood of 0.5 of being in the dictionary?

Now rather than estimating or computing the values of the θμs, the next step is to flip the variables in the distribution we seek, to P(s|theta), and then integrate out the thetas, based (it seems?) only on their prior distributions, Equation (5).

Q3.7: Now that s can again take value of either 0 or 1, why did we not need the distribution over θs in the case that s = 0? How can a distribution over s on the LHS in Equation (5) limit itself to the case of s = 1 on the RHS?

The calculations in the online methods may resolve some of these issues? But these all appear to start from Equation (5) and at least for me at this stage, the route to this equation is unclear.

*Reviewer #3:*

The manuscript suggests a new approach to learning coding dictionaries. The method is applied to neural population data recorded in singing birds. While the idea of the approach is interesting, it was hard to understand the details or how well it works. It seems that too much has been pushed to the SI, and more controls and examples over synthetic data sets would be useful. Moreover, and importantly – it was hard to interpret the results. Also, the current approach was not compared against other methods presented in recent years for learning metrics on neural population patterns, or even more simple clustering approaches.

Briefly, the authors use a log-linear model for the population words of N neurons and two behavioral states – using 2^(N+1) terms that span all potential orders of interactions among cells (theta's). To avoid arbitrary cutoff on the order of interactions that they consider, and use the important ones, they use a constructed prior distribution over theta's to learn a posterior model over the population words after integrating out the theta's. This gives an Ising model over 2^N interacting 'spins' where each sping now stands for a word that may or may not have appeared in the sampled data. The local fields acting on each of these spins is related to the prob of appearance of the words the spins stand for, and the pairwise interaction terms stand for an overlap between the words. These interactions now imply a 'competition' between words in terms of the explanatory power they carry (and are therefore typically negative)

Thus, if I understand this correctly, the constructed model gives a form of compression of the population 'codebook' by picking words that are dominant to be included in the dictionary and omitting similar words with less explanatory power and frequency.

Critically, it was unclear to me how one should evaluate the results. It is hard to interpret the structure of the dictionary in Figures4-5, and in particular over the pooling of conditions in Figure 5. In addition, there have been different approaches towards learning neural dictionaries in recent years, which are not mentioned here or compared against the current approach. Maybe this is too naïve, but it seems natural to ask how well would naïve clustering or Edit-distance based methods work here? Then, how does the current approach compare to more structured metrics such as those presented in Ganmor et al. *eLife* 2015, Prentice et al. Plos Comp Biol 2016, Rubin et al. Nature Comm, 2019, or Chaudhuri et al. Nature Neuro, 2019 ?

Finally, how would approach work for larger populations, where all observed patterns would appear just once?

It was not clear how should one interpret the nature of the organization of patterns in Figure 4B (the gap between lines, their very linear dependency?)

[Editors’ note: further revisions were suggested prior to acceptance, as described below.]

Thank you for submitting your article "Unsupervised Bayesian Ising Approximation for decoding neural activity and other biological dictionaries" for consideration by *eLife*. Your article has been reviewed by 2 peer reviewers, and the evaluation has been overseen by a Reviewing Editor and Aleksandra Walczak as the Senior Editor. The reviewers have opted to remain anonymous.

Essential revisions:

1) The numerical comparison to other existing methods is minimal and should be expanded. The analysis of the irreducibility of the codewords has insufficient support based on the numerical simulations. Moreover, the generality of the tool and comparison to other methods are discussed in almost entirely theoretical terms, which makes the claim on immediate utility for other datasets less convincing, especially outside the neuroscience community. In particular, the identified codewords are relatively short (3rd, 4th order stats) and N=20 is easy to fit an Ising model; so one would think that it would be relatively straightforward to fit the Ganmor model jointly. At least on artificial data one should be able to explore a larger N regime where there is a more immediate gain relative to existing models.

2) Although the paper is written as a methods paper, emphasizing the technical contributions and promising wide applicability to a range of different types of datasets, the numerical validation of the method is very much restricted to the statistical regime of the songbird dataset. From the perspective of a potential future user of the tool it's less clear how the method would behave on different datasets, and what needs to happen in practice for adopting the tool to data with different statistics. Based on current content, it would be better to focus the abstract and introduction on applications that are actually studied here.

3) The songbird analysis already reveals some challenges with respect to interpretability: in particular it is not clear how much information about the underlying neural processes can be revealed by summary statistics generated by the method, such as the number of codewords and their length distribution.

4) Limiting the analysis to N=20 patterns with somewhat random code structure makes it hard for a potential user of the tool to work out what needs to change when switching to a novel dataset (with respect to N, M, codeword complexity, sparsity of neural responses). Some work could be invested in spelling out the considerations of applicability with respect to quantities that an experimentalist would know about the data.

5) Is it possible to include multiple binary quantifications of behavior, similarly to how words are constructed from neural spike trains? For example, one can envision describing a particular song segment with respect to multiple binary features simultaneously.

*Reviewer #1 (Recommendations for the authors):*

Line 168: I think the notation here should P(s_mu=0) not s_mu=-1.

*Reviewer #2 (Recommendations for the authors):*

I am overall very excited about the framework and I see significant technical potential in the approach. Nonetheless, there are several missed opportunities in the numerical analysis, and things that could and should probably be improved to maximize the impact of the work on the broader community.

– Limiting the analysis to N=20 patterns with somewhat random code structure makes it hard for a potential user of the tool to work out what needs to change when switching to a novel dataset (with respect to N, M, codeword complexity, sparsity of neural responses). Some work could be invested in spelling out the considerations of applicability with respect to quantities that an experimentalist would know about the data.

– The codeword irreducibility could be analyzed in more detail, again in terms of experimentally relevant quantities.

– The comparison to other methods is minimal. The identified codewords are relatively short (third, fourth order stats) and N=20 is easy to fit an Ising model; so I'd think that it would be relatively straightforward to fit the Ganmor model jointly. At least on artificial data one should be able to explore a larger N regime where there is a more immediate gain relative to existing models.

[Editors’ note: further revisions were suggested prior to acceptance, as described below.]

Thank you for resubmitting your work entitled "Unsupervised Bayesian Ising Approximation for decoding neural activity and other biological dictionaries" for further consideration by *eLife*. Your revised article has been evaluated by Aleksandra Walczak (Senior Editor) and a Reviewing Editor.

The manuscript has been improved but there are some remaining issues that need to be addressed, as outlined below:

Essential revisions:

Thank you for the revised manuscript. The reviewers found the manuscript much improved and close to acceptance. Nevertheless, reviewers are suggesting a few additional analyses to demonstrate the advantages of the method and increase its impact.

1) Please consider the null model that includes pairwise correlations.

2) Please test the method to a larger dataset.

*Reviewer #2 (Recommendations for the authors):*

I generally find that the revisions have improved the overall clarity of the text, and addressed most of my questions.

Nonetheless, I found the answer to two of my concerns underwhelming:

– Discussing the practical considerations about translating the approach to other datasets (new discussion paragraph) is quite vague, essentially saying that the hyperparameters will more or less have to be figured out de novo. I was expecting a quantitative description of how different statistical properties of the data affect the process, which would translate into something closer to a concrete recipe for hyperparameters adjustment. The fact that this is not straightforward to do makes a broad adoption of the tool less likely.

– The concern about the overall interpretability of the extracted statistics remains unaddressed. I don't agree with the position is that "it only matters to estimate these statistics well, interpretation can come later." Being able to estimate a quantity does not necessarily make it useful to do. In the context of songbird specifically, the paper make the case for the results being difficult to interpret even when the statistics are well estimated. This seems like a pretty big problem, at least big enough to warrant a minimum amount of thought and a couple of sentences in the discussion.

*Reviewer #3 (Recommendations for the authors):*

The authors already made a great effort to improve their manuscript. However, from my side, I provide the following concerns about the significance and broad impact of the current revised manuscript.

Strength of the manuscript

The traditional model of maximum entropy type requires a large amount of data to predict effective interactions among elements (e.g., neurons), and fails to identify statistically over- or under-represented (relative to a null model) codewords, and thus the neural-behavior prediction remains elusive. This manuscript provides a novel approach that can systematically search for the significant codewords, yet requiring fewer data samples (compared to the number of higher-order model parameters).

The method first writes all orders of interactions in a probability distribution form; finding non-zero higher order interaction parameters could tell which codewords should be included in the neural dictionary and how they matter. To make the method algorithmically solvable, the authors turn the optimization into an Ising model of interacting indicator variables. This indicator variable is responsible for identification of key codewords. The inferred value of the external fields signals the importance of the codeword, while the sign of the high-order interaction parameter reflects whether the word is over or under-represented. To explain the neural-behavior correlates, the authors also include the behavior variable into the components of the activity. This framework works consistently in the songbird motor experiments.

Weakness of the manuscript

To relate the neural activity and behavior, the authors test their method in both synthetic data and songbird real data. There are two weakness, considering the impact of the method in a broad context. First, the variable size is limited to N~20, which is severely below the number of neurons the neural data scientists get interested in (e.g., for modeling retinal ganglion cells, and even cortical cells). This may be a tradeoff between all order of interactions to be considered and the number of model parameters. Second, a null model is chosen to be an independent model that neglects correlations in the data, which may affect the characterization of the redundancy of the uBIA model. Even if an irreducible ensemble of codewords is obtained, what is the predictive power of this ensemble with respect to control the macroscopic behavior (e.g., skilled motor control or sensorimotor learning), rather than showing statistics of neural-behavior activity, remains obscure in the current manuscript.

Considering the strength and weakness of the manuscript, I recommend the following issues for the authors to improve their idea.

1. The proposed model is limited to symmetric couplings and i.i.d data samples. First, in a neural circuit, the asymmetric coupling is common among neurons; Second, the temporal order of each spiking activity may play a key role in encoding information. For example, a paper "Blindfold learning of an accurate neural metric" (PNAS, 2018) takes the temporal information into account. When putting in the context of neural-behavior relationship, these two factors would become important. But they seem neglected in the current manuscript.

2. To achieve a higher-order-interaction-included and non-redundant model is really important problem in the field. For example, the reliable interaction model proposed in ref. 26 is not fully compared in the current manuscript, i.e., what is the new advance? This is better to clearly demonstrated in a plot in the main text. Second, a recent paper (Phys Rev E, 104, 024407, 2021) used an information-based criterion to identify statistically significant couplings, which may be applicable to the context the authors consider here. In this *eLife* submission, the method relies on the magnetization inclusion threshold, which may be expensive for real data analysis.

3. On the equation 1, the authors seem to use the local model parameter to detect the global significance of the collective codeword, which I could not understand very well, because a codeword is determined by all order of interactions considered in the log-linear model.

4. Technically, the logic going from Equation 7 to Equation 8 is broken, since the s-dependence in Equation 8 does not naturally arise from Equation7.

5. I could not understand well how the sign of reflects whether the word is over- or under-represented, and even, how the parameter account for the reducibility of the dictionaries. This part is better to be expanded in the manuscript, although these parameters have highly non-linear function relationship.

---

## [Author Response]

[Editors’ note: the authors resubmitted a revised version of the paper for consideration. What follows is the authors’ response to the first round of review.]

The individual reviewer comments are appended below and can be used either as directives for that new submission, or as helpful pointers for a submission elsewhere. Summary thoughts are included here, with the detailed reviews from the 2 reviewers included at the end.The paper details a new approach to learning the input/output relationships in complex biological systems that drive things like motor behavior or (perhaps) protein folding and other processes. The work focuses on birdsong as an example of a complex motor task potentially modulated by precise temporal patterns of spiking in the bird motor cortex (RA). The method claims to be able to use very few jointly recorded samples of the input and output to learn both a model for the distribution of the spiking patterns observed as well as how they drive behavior. The method uses strong priors in a Bayesian Ising Approximation to identify the neural code words that are expressed at unusually high or low frequency with respect to the behavioral output. The authors find a large number of spike patterns correlated with behavioral variations in syllable pitch. The method has potential applications beyond birdsong; the ability to fit input/output relationships with very few samples is appealing.

This is an excellent overall summary of the paper. However, there is one crucial aspect of our approach which is summarized incorrectly and which we clarify briey here as well as in detail in the revised manuscript: uBIA does not fit or reconstruct the input/output relationship from data. Rather, it detects which keywords contribute statistically significantly to the input/output dictionary, but unlike many other methods, it does not build a model of the input/output relationship. This is crucial, since it is one of the reasons why comparing uBIA to other approaches is hard, and it is also one of the main reasons why uBIA can work with fewer assumptions and smaller datasets compared to other methods. We have emphasized that uBIA does not build an explicit model of the data in the new revision.

The reviewing team, however, had some major reservations about the results and felt that the current draft did not give them enough details to properly evaluate the method. A summary of those major points is given here:1) The method:The reviewers felt that there wasn't enough detail presented to clearly analyze the results of the model. Much of the detail was in the SI and Methods, where the exposition wasn't clear or detailed enough for the reader to properly assess the results.

Following these concerns, we have completely restructured the paper, moved some of the calculations from the Methods into the main text, de-emphasized biological applications and re-emphasized and explained in more detail the algorithmic details, as described below.

Questions raised included:– What's the null model?

In the revised manuscript, we now explicitly define the null model after Equation 3.

– How does this method scale to larger neural populations?

It is hard to be precise answering this. The size of the population is not the most crucial variable since, as is, the method analyzes 2*^N^* possible terms in a model with just *N* variables. What is more important is the quality and quantity of sampling, the strength of the contribution of terms to the likelihood, and so on. We discussed this explicitly in the Discussion section, but also throughout the manuscript, wherever relevant, such as in the first paragraph of Page 4 and in the last paragraph of the “Testing and fine-tuning uBIA on synthetic data” section.

– Is the clustering used here justified? Can it be compared to other, simpler clustering methods?

We apologize, but we do not understand the question here, since there is no clustering used in our method. We hope that the other revisions and clarifications described here are sufficient to clarify our overall approach.

– How do you justify the choice of prior and how are the unknowns fit from the data?

We have completely reworked the presentation of the method, and we hope that the new section “The unsupervised BIA method (uBIA) for dictionary reconstruction” answers these questions. Briefly, as explained in this section, we choose the priors we work with because they are standard (e.g., Gaussian), and allow for the strongly regularized regime, which is a key for our work. And then we chose the weakest of such strongly regularizing priors (largest), as detailed in the manuscript.

– Other extant methods for finding reliable interactions in exponential models were not referenced or compared.

We now introduce the necessary references and comparisons, largely in the first section of the supplementary materials. Briefly, as we discussed in the preamble to this response letter, the goal of our method is very different from many others. Thus, as explained in the revised Supplemental Information, our method is complimentary to, rather than a competitor of, the other methods we review, which complicates the task of comparing them.

– Artificial data used in the SI are based on properties of the recorded neurons, and as the authors explain the cells seem to be quite weakly coupled, so the relevance of the model for other data is not clear.

Just to make sure there is no confusion: the data is from single cell recordings, and the interactions are between time points of neural activity, and not within a population of neurons. However, our response to the substance of the comment is that there is no way a priori to know if a method would work for other data, without trying it. We explain this in the revised Discussion section, where we also comment on what the determinants of the success will be. We only have access to the neural dataset, and analyzing it was a lot of work – we simply cannot do more in one article. To check if the method works in other fields, we must first publish it, so that other researchers can use it on their data. This is precisely why we are submitting the article to *eLife*.

2) The behavioral results:The reviewers were not convinced by the pitch results, primarily because of the coarse-graining of pitch into just two categories. Again, much of the explanation was relegated to the Methods part of the paper and could have been presented more clearly. If this method works, this is clearly interesting, but the results presented seemed rather weak.

We thank the reviewers for directing our attention to this. As the reviewers rightly point out, uBIA uses discretized measures of motor output, and in some cases such discretization will discard relevant behavioral data, as when a (continuous) measure of vocal pitch is discretized into two categories. However, as explained below this coarse-graining does not limit the usefulness of the uBIA approach, either in the specific case of the songbird vocal motor system or in its usefulness to biologists more generally. In the case of the songbird data, Dr. Nemenman and Sober’s prior work (Tang et al., 2014), which analyzes the same dataset, has demonstrated significant mutual information between behavior and millisecond-scale spike timing in neurons of the songbird motor cortex, even when the continuous pitch variable was discretized into just two categories. Moreover, that same paper demonstrated similar statistical dependencies when pitch was discretized into 3, 5, or 8 groups, showing that information between millisecond-scale (cortical) spike patterns and a coarse-grained representation of behavior does not depend critically on the details of the discretization.

More generally, the loss of resolution due to the (artificial) discretization is inevitable. However, it would be hard for the discretization to create new codewords – by discretizing, we largely lose codewords, for which the behavior is mapped into a single bin. We believe that this is acceptable – the method does not claim to discover all codewords (this would require infinitely large datasets!), and so, as long as we keep making the same error again and again (deliberately missing some possible keywords), and still come out with relatively large dictionaries reconstructed, the approach remains useful. In addition, one can partially address this concern by trying different binarizations. Specifically, in the last part of the paper, we did just that – binarizing the data into different groups, and hence exploring the exploration-exploitation tradeoff. We now emphasize this in the text, in the header of the “Reconstructing songbird neural motor codes” section and at the end of the “Dictionaries for exploratory vs. typical behaviors” section.

3) The format of the paper:The paper read a bit like 3 manuscripts glued together, each of which has merits, but none of which seemed to satisfactorily stand on its own. The paper starts with a grand introduction that talks about solving a major challenge in quantitative biology. Then it quickly dives into something that reads more like a methods paper, but with insufficient exposition and clarity. Finally, the paper shifts to the application of the method to behavior and neural data, the results of which seemed somewhat weak and preliminary.

We agree with these concerns, and we took them seriously. To respond to them, we have completely changed the paper structure, and the paper is now a methods paper. We think the exposition has improved because of the change, and we hope that the reviewers agree.

Taking these critiques together, the reviewing team felt that the paper could not be reasonably revised for eLife as a Research Article. However, the team did find the central method potentially interesting if more details are provided that allow for proper assessment of it. The team felt that it could be a good "Tools and Resources" advance, if the method holds up to this more detailed scrutiny.

Thank you. This is a great suggestion, and we are resubmitting the paper as a “Tools and Resources” article here.

Finally, we also change Figure 4 to make it easier to read.

Reviewer 2:The paper is a potentially interesting application of a recent statistical method, the Bayesian Ising Approximation (Fisher and Mehta), aimed at addressing the problem of estimating a complex distribution, in particular the relationship between a code and an output, from very few samples. This is accomplished by heavily weighting by the prior. The authors have applied this method to support the idea that there are precisely timed spike patterns that are especially significant in the generation of behavior, in this case in the output of syllables of birdsong. They use the method to identify "code words", patterns that are claimed to appear with notably high or low frequency in correspondence with a behavioral variable. They apply this method to find a larger number of precise spike patterns correlated with behavioral variations in the tails of the behavioral distribution. They suggest an interpretation that perhaps larger exploratory motor gestures require a more precise spiking code; although it might also be argued that this shows that there are more ways to achieve these larger fluctuations? While the claim is counterintuitive, it rests on the power of the method, whose exposition left a number of gaps.1. It is unclear to what extent this goal is meaningful in the case that the paper analyses: premotor drive of muscles in the birdsong pathway. It seems evident that timing of RA spikes necessarily sculpt motor output; and the authors have previously analyzed this in detail to reveal timing precision (Tang et al.). It was not clear, however, that specific patterns meaningfully emerge from a sea of similar spike trains to encode exact outputs. The authors ask whether precise patterns in 2ms bins have a significant relationship with binarized song characteristics in 40ms bins-a very low fidelity relationship on the "behavior" side compared with a high level of precision on the neural side. This seems a dramatic discrepancy between the dimensionality of neural response vs behavior that deserves more justification. How general a space of coding relationships (nonmonotonic, etc) does this method explore given this binarization?

We answered this question in detail above (see “(2) The behavioral results” in our reply to the Editors above). Briefly, by discretizing, we certainly lose a lot of possible coding relationships – but we also lose many of them simply due to the finite size of the data sets. These are the errors that everyone working in the field has to agree to make: it is ok to lose codewords, as long as we do not introduce many spurious ones. Finally, exploring different discretizations certainly increases the diversity of the codes we can detect – and we do, indeed, detect nonmonotonic codes in the section on exploratory vs. typical behavior.

2. Further, it was unclear whether a temporally structured firing rate would be equally convincing as a representation of behavioral variation. While the concept of "irreducible dictionaries" is appealing, does this method fairly compare a timing code to a rate code? For example, doesn't the competition component of the model suppress the identification of firing rate-like contributions? Looking at Figure 1, the cartoon of eliminating overlapping but similar code words seem equivalent to ignoring a less temporally precise method of coding for behavior. That is to say, if you are forcing your model to choose one of several similar code words in a winner-take all inclusion choice, isn't the timing precision you see partially a factor of not looking at the other similar code words which didn't win out?

Thank you for this important question. Parenthetically, our method actually allows for partially overlapping words, as the dictionaries shown by tic marks in Figure 5 illustrate. However, our main response to this comment is that the framing of the time-rate debate by the reviewer is inconsistent with our view of the matter. Of course, both the rate and the precise timing contribute to encoding information about the ensuing behavior – the question is how much is contributed by each, and how. As we tried to elucidate in our recent review paper (Sober et al., 2018), we view neural codes on a two-dimensional continuum, where temporal precision is on one axis, and the order of the code is on the other (see Figure 2d of Sober et al., 2018 for a reproduction of the argument from the above mentioned review). While allowing the rate code to change the rate over orders of magnitude on a millisecond scale would introduce the millisecond-level precision, it would still be a first order code, not requiring to know the neural activity at two instances to predict the behavior. To the extent that most of the codewords that we observe involve multiple spikes, cf Figure 5c, the codes are not simple first order rate codes – they are pattern codes, with temporal resolution of a few ms. We have checked and correct the entire manuscript so that nowhere in the manuscript we contrast timing to rate, but instead we focus on the difference between rate and timed patterns code.

The comparison of this method to GLM methods of relating the spiking to behavior is unclear. One alternative model was analyzed in the online methods, but is not clearly explained and such comparisons deserve to be treated in the main text. It should also be easy to report the simple 0th order analysis of how well you can predict the binary version of behavior simply with a spike count within the 40 ms window, or with some intermediate level of temporal variation. Seeing these comparisons as a control of the assertion that precise timing matters in this system would enhance intuition for the necessity of this method and better justify the claim "that they predict behavioral features better than other approaches" (line 256).

Thank you for this important question. As we showed in Tang et al. (and mentioned in the manuscript), there is essentially no information between the spike count at 20+ ms resolution and the pitch, which makes making such comparisons not illustrative. Crucially, as we now stress throughout the manuscript, and in particular in “Overview of prior related methods in the literature” (see also our response to the introduction of the Editorial Decision letter), uBIA does not build a generative model of the underlying distribution; it merely detects which features need to be part of the model, and after that one can build the model using any number of other methods, including GLMs. In other words, uBIA doesn’t compete with (and should not be compared to) other methods, and should be viewed as a complementary technique.

3. In general, the exposition of the method leaves many questions and could be improved considerably in clarity. I have tried to summarize the logic of the paper's core argument below and underscored throughout where more help could be given to understand the meaning and generality of assumptions made. Logic of the calculation:

We agree that the original manuscript has left a lot to be desired, and we hope that the extensive changes we have made to the manuscript have solved these problems.

The paper sets up the problem of writing down a model for the joint probability of a behavioral event s0 and the firing of a spike in each, i, of N bins, σi. The authors write this in terms of an expansion over every possible combination of σi, Log P(σ|θμ) = -log Z + Σ θμ Π _{i in Vμ} σi.Q3.1: Please expand a little on the exponential form. In maximum entropy approaches, this form of distribution arises from the maximum entropy solution, but if we are avoiding MaxEnt approaches, can you briefly justify it?

This is the most general form of the underlying probability distribution, and it does not lead to any loss of generality. We now emphasized this in the text before Equation (1).

In this approach, a firing pattern with spikes in bins (1,2,3,4) is considered to be independent of the events (1,2) and (3,4); whereas in maxent, if one is only expanding up to order 2, one hypothesizes that the probability of the word (1,2,3,4) will be absorbed into the pairwise probabilities, and one can ask how different the data is from that guess. It seems that here one is parametrizing with vastly more parameters-one for every possible spike/behavior combination-- without a systematic method of cutting them down (or at least one that was clear to me; it is addressed at the end of the Online Methods calculations once we have moved into the s variables but it would be helpful to get some intuition from the start).

This is not quite correct – the whole point of the method is that one should (and does) consider all possible combinations, but then one systematically evaluates the evidence for needing to include a certain combination into the probability distribution. We hope that the current revision, which has moved the appendix into the main text, has resolved this ambiguity. Specifically, the paragraph before Equation (2) has been revised to address this question as well.

One then defines "indicator variables" sμ which tell us which patterns appear significantly less or more often than one would expect under some null model. It would be good to specify this null model right away, and in straightforward terms; I don't see it defined in the main text at all.Q3.2: Please explain at this point the null model for the frequency of appearance of any possible pattern of spikes and its correlation with behavior? Should it not depend on the specific behavior?

In the revised manuscript, we now explicitly define the null model after Equation 3. Briefly, the null model matches the firing rates in all bins to experimental observations.

It is unclear why one is now allowed to write that the original definition of the model is EQUAL to the model now written including only terms defined by a criterion defined by our subjective interest in them: Log P(σ|θ) = -log Z + Σ θμ sμ Π _{i in Vμ} σiQ3.3: Why does one not here need to define a new and different probability distribution, something that makes some claim about belonging to the codeword dictionary, in which these terms are the only ones that appear?

Thank you very much for this critique. Indeed, what’s written in Equation (2) is the same distribution as in Equation (1), following a series of algebraic transformations. We now stressed it in the text before Equation (2) that no new quantities are being defined.

Now the authors turn to the task of estimating the 2^N parameters θμ. Instead of doing this estimation directly from the data (i.e. the a posteriori expectation), they use a prior on these values: they choose to assume that each of the values are narrowly distributed about some mean θ*μ: P(θ*μ : sμ=1) ~ exp(-(θμ −θ*μ)2). This distribution is taken only to hold if the word is indeed a code word, ie statistically associated with a behavior. If it is not a code word, the prior is not defined.Q3.4: Please justify this choice in more detail. Why are the epsilons not dependent on μ?

It would be impossible to obtain a simple a posteriori expectation here – with 2*^N^* equal to about 2*,*000*,*000 in our example, and with only a few hundred samples, the uncertainties would be too large. This is where Bayesian methods are useful and needed. We explained in the revision why we made this specific choice of the prior. We would like to point out here that we do not define the prior for non-codewords simply because it doesn’t matter – those words won’t contribute to the distribution at all.

Q3.5: The argument for where these 2^N values of θ*μ come from is very unclear. Please spell out more clearly how constraining the means of the individual σis provides sufficient constraints for all 2^N values, or why this is not needed.

There are 2*^N^* values of *θ*s because there are 2*^N^* different inclusion/ exclusion combinations of *N* spins. The constraining on the rates (means) does not provide sufficient information for all 2*^N^*. This is, in fact, the point of the algorithm: there’s no way to determine 2*^N^* values from small data sets, and we thus need to do something else. Specifically, we only ask a question of which *θ*s are statistically significantly nonzero, and hence should be included as codewords, but we never try to determine their values directly. We hope that the current revision makes this point clear.

Q3.6: Why does a code word have an a priori likelihood of 0.5 of being in the dictionary?

As in any Bayesian analysis, we need to define a prior, and we choose the least informative 50/50 prior in our case – a priori we do not know whether a particular codeword is used or not used by the animal. If we had a reason to believe a priori that a certain word should or should not be included in the dictionary with a different probability, then that different prior could be used. We now explain this in the text in the paragraph following Equation (3).

Now rather than estimating or computing the values of the θμs, the next step is to flip the variables in the distribution we seek, to P(s|theta), and then integrate out the thetas, based (it seems?) only on their prior distributions, Equation (5).

This is not entirely correct. We are interested in the posterior distribution of *s* given the observed data, and integrating out *θ*. We choose to integrate *θ* out since, as we argued before this equation, the posterior expectation of *θ* cannot be determined with a reasonable accuracy from datasets of realistic sizes. Here, crucially, the a priori expectations an*d* data get combined, and the integration is not controlled just by the prior distributions.

Q3.7: Now that s can again take value of either 0 or 1, why did we not need the distribution over θs in the case that s = 0? How can a distribution over s on the LHS in Equation (5) limit itself to the case of s = 1 on the RHS?

It’s not that we do not need the distribution of *θ*s for *s* = 0, it’s that the distribution does not matter: if *s_µ_* = 0, the corresponding *θ_µ_* has no effect on the distributions of data. We hope that this has become clearer in the current revision.

The calculations in the online methods may resolve some of these issues? But these all appear to start from Equation (5) and at least for me at this stage, the route to this equation is unclear.

We hope that this has become clearer in the current revision, which included a major restructuring of the paper.

Reviewer #3:The manuscript suggests a new approach to learning coding dictionaries. The method is applied to neural population data recorded in singing birds. While the idea of the approach is interesting, it was hard to understand the details or how well it works. It seems that too much has been pushed to the SI, and more controls and examples over synthetic data sets would be useful. Moreover, and importantly – it was hard to interpret the results. Also, the current approach was not compared against other methods presented in recent years for learning metrics on neural population patterns, or even more simple clustering approaches.

We have substantially revised this manuscript to address this concern, and discuss this in the “Overview of prior related methods in the literature” section, as well as in the reply to the editorial decision above. In particular, we added a description emphasizing that our method does not compete with most others in the literature, and cannot really be compared to them, as the goals are different. Indeed, uBIA learns dictionaries – in other words, it detects which words are codewords. Most other methods, in contrast, take a predefined set of putative codewords, and then build a generative model based on them. While usually the set of putative codewords is defined as words with some number of spikes, nothing prevents to use these other methods in conjunction with uBIA, so that uBIA first detects the codewords, and other methods are then used to build a model based on them.

Briefly, the authors use a log-linear model for the population words of N neurons and two behavioral states – using 2^(N+1) terms that span all potential orders of interactions among cells (theta's). To avoid arbitrary cutoff on the order of interactions that they consider, and use the important ones, they use a constructed prior distribution over theta's to learn a posterior model over the population words after integrating out the theta's. This gives an Ising model over 2^N interacting 'spins' where each sping now stands for a word that may or may not have appeared in the sampled data. The local fields acting on each of these spins is related to the prob of appearance of the words the spins stand for, and the pairwise interaction terms stand for an overlap between the words. These interactions now imply a 'competition' between words in terms of the explanatory power they carry (and are therefore typically negative).Thus, if I understand this correctly, the constructed model gives a form of compression of the population 'codebook' by picking words that are dominant to be included in the dictionary and omitting similar words with less explanatory power and frequency.

This is, indeed, a reasonable summary of our paper. We would like to point out, however, that compression is not the primary goal here, and it emerges naturally, not at the expense of the quality of the fit to the data. In other words, if there is sufficient evidence for even overlapping words to be included in the dictionary, then both will be. Finally, some words – especially non-overlapping ones – may have positive interactions, promoting rather than inhibiting each other.

Critically, it was unclear to me how one should evaluate the results. It is hard to interpret the structure of the dictionary in Figures4-5, and in particular over the pooling of conditions in Figure 5.

We thank Reviewer for these comments. Some of them were answered above in our response to the editorial summary questions, and we refer the Reviewer there, to our response to “The behavioral results” editorial comments. In addition, here we point out a few more things. First, consistent with our resubmitting this paper as a “Tools and Resources” contribution rather than a research article, we have de-emphasized the interpretation of the birdsong data and focused on the mechanics the method itself. Second, it is simply not possible to do give a full interpretation requested by the Reviewer, without pooling, since we lack tools to collect the amounts and the types of data we would need. In particular, pooling is done, in part, to overcome the statistical power problems caused by small datasets. Further, the ultimate test of any method for dictionary reconstruction must be causal stimulation: one needs to build a model based on the detected codewords, predict responses to possible experimental perturbations, and then apply the perturbation and compare the results to the predictions. We do not do this here, but neither do the absolute majority of other publications in the field (our previous publication on bird breathing is one of the few exceptions, and the goal is to do something similar in the current system as well, but this requires development of new experimental methods). We already did what the best publications in the field do – we build the dictionaries, and then used them, in the “Verification of the inferred dictionaries”, section to build a simple statistical model of the code, and we showed that this model outperforms at least some competitors. Beyond this, we do not know how to provide a more useful biological interpretation. However, again, the same concern is applicable to essentially any other publication in the field.

As argued in the previous paragraph, pooling of conditions in Figure 5 (now Figure 6), indeed, adds an additional complexity to interpretation of individual dictionaries, as using a pooled dictionary to predicted pooled data would certainly be pointless (this would be like trying to translate from a corpus of books written in different languages using a mixture of dictionaries that translate between different pairs of languages). However, what would make more sense in such pooled books analogy – and is precisely what we do for the birdsong data – is to ask questions of the type: how are Romance languages different from Germanic ones? Are the words of similar length? Are roots similar? etc. We hope that this analogy is illustrative for the Reviewer.

In addition, there have been different approaches towards learning neural dictionaries in recent years, which are not mentioned here or compared against the current approach. Maybe this is too naïve, but it seems natural to ask how well would naïve clustering or Edit-distance based methods work here? Then, how does the current approach compare to more structured metrics such as those presented in Ganmor et al. eLife 2015, Prentice et al. Plos Comp Biol 2016, Rubin et al. Nature Comm, 2019, or Chaudhuri et al. Nature Neuro, 2019?

We now review the relation of all of the methods mentioned by the reviewer to uBIA in “Overview of prior related methods in the literature”. Here we also add that, as we now emphasize in the manuscript, our method is complementary to the ones mentioned by the reviewer, and does not compete with them. These methods can be used to reconstruct models of the neural code *after* uBIA detects which keywords should enter the model.

Finally, how would approach work for larger populations, where all observed patterns would appear just once?

Performance of uBIA will certainly degrade in this situation. However, no method will work well then, so that this cannot be viewed as a something particularly bad about uBIA. That said, uBIA has two major strengths in this regime, which are now emphasized in the manuscript. First, uBIA analyzes not just patterns, but all sub-patterns. In other words, as long as there are even partial coincidences among the spiking patterns, they will be detected, and these coinciding sub-patterns will become keywords in the dictionary. It is clear that at least some short sub-patterns must coincide in any reasonable dataset – e.g., there are only *N* first order words formed by *N* interacting units – and any realistic dataset will have, at least, Μ»Ν, ensuring that, at least, such simple one-spike (and maybe even many two-spikes) words have good statistics behind them. Second, uBIA detects not just over-representation of patterns, but also their under-representation. Thus, if a complex spike word happens once (or does not happen at all), while it should happen many times based on its composition from small sub-patterns, such word could enter the uBIA dictionary as well, with its absence potentially encoding the behavior.

It was not clear how should one interpret the nature of the organization of patterns in Figure 4B (the gap between lines, their very linear dependency?)

In Figure 4B (now 5B), the reason that there are no points near *x* = *y* (in other words, there is a gap there) is because *x* = *y* means that the observed frequency of a spike codeword is roughly the same as its frequency expected from the observation of its various subparts. Then, by definition, it isn’t significantly under/overrepresented. The width of the gap around the diagonal is determined by various significance threshold, which we set as explained in the text.

[Editors’ note: what follows is the authors’ response to the second round of review.]

Essential revisions:1) The numerical comparison to other existing methods is minimal and should be expanded.

We have argued in our previous submission that there really are no other methods to compare to, designed to work in the regime similar to uBIA. It seemed to us that it would be unfair to run other methods on our datasets, see them not work well (as expected – because they make assumptions that are invalid in our regime), and then claim success. However, since the concern has been raised again, we really have to address it. To do this, we added a section in the *Online Methods* “Direct application of MaxEnt methods to synthetic and experimental data”, in which we compare uBIA to the relevant interactions model of Ganmor et al., with which uBIA has the highest similarity. The results are as expected – a method not designed for our data regime fails. We emphasize here again that the relative superiority of uBIA on these data should not be taken as a slight directed at other methods, but rather as an indication than, to cover different data regimes, multiple methods should be combined. We emphasized this in the “Overview of prior related methods in the literature” supplemental section.

The analysis of the irreducibility of the codewords has insufficient support based on the numerical simulations. Moreover, the generality of the tool and comparison to other methods are discussed in almost entirely theoretical terms, which makes the claim on immediate utility for other datasets less convincing, especially outside the neuroscience community. In particular, the identified codewords are relatively short (3rd, 4th order stats) and N=20 is easy to fit an Ising model; so one would think that it would be relatively straightforward to fit the Ganmor model jointly.

We hope that the addition of the new comparison figure (described above) partially alleviates these concerns. Additionally, we point out that 3rd and 4th order words are long, as most others deal with just pairs, as illustrated in the new Figure 7. Indeed, it is not easy to fit an *N* = 20 Ising model with 4th order terms, because there are 20 ∗ 19 ∗ 18 ∗ 17*/*(4 ∗ 3 ∗ 2 ∗ 1) = 4845 terms in this model, which cannot be fit from just a few hundred samples, which is precisely why the Ganmor model fails in this case (Figure 7).

At least on artificial data one should be able to explore a larger N regime where there is a more immediate gain relative to existing models.

Indeed, in the newly-added Figure 7 we explored dataset sizes up to *M* = 10^5^, showing that uBIA begins detecting interactions with datasets orders of magnitude smaller than those required by alternative methods.

2) Although the paper is written as a methods paper, emphasizing the technical contributions and promising wide applicability to a range of different types of datasets, the numerical validation of the method is very much restricted to the statistical regime of the songbird dataset. From the perspective of a potential future user of the tool it's less clear how the method would behave on different datasets, and what needs to happen in practice for adopting the tool to data with different statistics. Based on current content, it would be better to focus the abstract and introduction on applications that are actually studied here.

We have edited second half of the abstract and a few sentences in the Introduction (see latexdiff file) to make it clear that our main applications to date have been to songbird data.

3) The songbird analysis already reveals some challenges with respect to interpretability: in particular it is not clear how much information about the underlying neural processes can be revealed by summary statistics generated by the method, such as the number of codewords and their length distribution.

The reviewer is correct that our analysis of the songbird data raises a number of important questions for future studies. Although these remain to be answered, we emphasize that before the biological interpretation of over/underrepresented neural patterns can be attempted, such patterns must first be identified. uBIA therefore represents a crucial advance in our ability to address these questions.

4) Limiting the analysis to N=20 patterns with somewhat random code structure makes it hard for a potential user of the tool to work out what needs to change when switching to a novel dataset (with respect to N, M, codeword complexity, sparsity of neural responses). Some work could be invested in spelling out the considerations of applicability with respect to quantities that an experimentalist would know about the data.

We agree with the reviewer and have added a discussion of how to set hyperparameters (second-to-last paragraph of the revised Discussion).

5) Is it possible to include multiple binary quantifications of behavior, similarly to how words are constructed from neural spike trains? For example, one can envision describing a particular song segment with respect to multiple binary features simultaneously.

We explicitly examine this question in “Dictionaries for exploratory vs. typical behaviors” and the corresponding Figure 6, which repeats our analysis for different binary discretizations of our behavioral data.

Reviewer #1 (Recommendations for the authors):Line 168: I think the notation here should P(s_mu=0) not s_mu=-1.

Thank you, this has been corrected

Reviewer #2 (Recommendations for the authors):I am overall very excited about the framework and I see significant technical potential in the approach. Nonetheless, there are several missed opportunities in the numerical analysis, and things that could and should probably be improved to maximize the impact of the work on the broader community.

We thank the Reviewer for their generally positive comments.

– Limiting the analysis to N=20 patterns with somewhat random code structure makes it hard for a potential user of the tool to work out what needs to change when switching to a novel dataset (with respect to N, M, codeword complexity, sparsity of neural responses). Some work could be invested in spelling out the considerations of applicability with respect to quantities that an experimentalist would know about the data.

Please see our response to this issue above under “Essential revisions.”

– The codeword irreducibility could be analyzed in more detail, again in terms of experimentally relevant quantities.

Please see our response to this issue above under “Essential revisions.”

– The comparison to other methods is minimal. The identified codewords are relatively short (third, fourth order stats) and N=20 is easy to fit an Ising model; so I'd think that it would be relatively straightforward to fit the Ganmor model jointly. At least on artificial data one should be able to explore a larger N regime where there is a more immediate gain relative to existing models.

Please see our response to this issue above under “Essential revisions.”

[Editors’ note: what follows is the authors’ response to the third round of review.]

Essential revisions:Thank you for the revised manuscript. The reviewers found the manuscript much improved and close to acceptance. Nevertheless, reviewers are suggesting a few additional analyses to demonstrate the advantages of the method and increase its impact.1) Please consider the null model that includes pairwise correlations.

This is not a simple request. For starters, most methods that aim to reconstruct features of multivariate probability distributions from data use the product of marginal distributions as the null model. In this context, our approach is consistent with the best practices. Further, if we were to follow this request, we think that this would create more issues than it would solve. In the regime we operate in, where the number of pairwise correlations is only a few times smaller than the dataset size, any method for constructing a null model that respects pairwise correlations in data would result in a null model that itself has substantial error bars. In our subsequent estimate of deviation of frequencies of various patterns from the null model, we would then be hampered by these large error bars, and whether some words should be or shouldn’t be included into the dictionary would depend on these fluctuations. To be sure, we agree that extending our method to a different null model is a worthy research project – but this is a research project that will take months of work to do right, and we would prefer to publish the current paper first, before studying such complicated extensions.

In summary, consistent with the second decision letter, we chose not to perform these additional analyses. However, in the revised Discussion, we now suggest that this expansion of the algorithm would be a worthy goal, but also argue briefly that it is not an easy task.

2) Please test the method to a larger dataset.

Again, while we appreciate where the comment is coming from, we would like to push back on this. Our understanding is that by “larger dataset”, the Reviewers mean larger *N*, the number of units, rather than *M*, the number of samples, and they are requesting us to list which specific changes to the algorithm’s parameters we need to adopt in this case. Our concern is that we already provided a recipe for parameter adjustment: count the number of patterns that are identified as predictive of both presence and absence of behaviors. Those are obviously wrong, and parameters should be set to avoid them. Any more specific suggestions will depend on not just the size of the dataset, but also on how many patterns really contribute to the underlying distribution, and what is the distribution of the coupling constants associated with these patterns. While doing some runs over the dataset size is relatively easy, we do not see this providing a substantial insight, since all of our findings would be specific to the statistical properties of the data we analyze. In other words, we would like to request the permission to publish the manuscript with no additional simulations of larger datasets. When published, we will then focus on adapting the methods to specific datasets of interest to us and others. For example, one such interesting dataset comes from recording multiple single motor units in marmosets. There *N* is naturally larger, so that we will be further developing our methods for a concrete application, rather than for solving toy models.

Nonetheless, we took this request very seriously. Now there are two additional paragraphs in the Discussion, which focus on how one would set the detection threshold and the inverse regularization strength, the two main hyperparameters of our algorithm.

Reviewer #2 (Recommendations for the authors):I generally find that the revisions have improved the overall clarity of the text, and addressed most of my questions.

Thank you! We are glad to hear this.

Nonetheless, I found the answer to two of my concerns underwhelming:– Discussing the practical considerations about translating the approach to other datasets (new discussion paragraph) is quite vague, essentially saying that the hyperparameters will more or less have to be figured out de novo. I was expecting a quantitative description of how different statistical properties of the data affect the process, which would translate into something closer to a concrete recipe for hyperparameters adjustment. The fact that this is not straightforward to do makes a broad adoption of the tool less likely.

As we tried to emphasize in the response to the Editors above, we do not expect there to be a practical guide at this point. We have analyzed many datasets in our careers, and we firmly believe that interesting discoveries require one to operate at the edge of applicability of any method, so that no standard, simple prescription would work. In our specific case, there are simply too many parameters to explore: the number of measurements *N*, the number of units *M*, the number and the order of the true words contributing to the dictionary, the statistics of the coupling constants associated with these words, and so on. We do not expect there to be simple heuristics, and the Reviewer is absolutely correct that one will have to work hard to adjust the algorithm to each new qualitatively different dataset.

The Reviewer is rightly worried that this may decrease the adoption of the algorithm, but we believe that this is not going to be the case. Indeed, as a part of the Simons-Emory International Consortium on Motor Control, we ran training workshops to disseminate the method in the community. The workshops were attended, cumulatively, by many dozens of trainees from within and outside the Consortium. That is, the method is being adopted by a number of labs. This has created a weird dynamics where these labs, in their own publications, want to refer to an article with the original description of the method, and this article (the current one!) is not yet published.

In other words, we would prefer to publish the manuscript now with minimal changes and explore different data regimes in future work. That said, we added a shortened version of the above arguments, and a succinct description of how one should set the hyperparameters of the algorithm to the Discussion section of the manuscript. Specifically, there are now two new paragraphs where we focus on setting correct values for the detection threshold *m* and the inverse regularization strength *ϵ*, the two most important hyperparameters of uBIA.

– The concern about the overall interpretability of the extracted statistics remains unaddressed. I don't agree with the position is that "it only matters to estimate these statistics well, interpretation can come later." Being able to estimate a quantity does not necessarily make it useful to do. In the context of songbird specifically, the paper make the case for the results being difficult to interpret even when the statistics are well estimated. This seems like a pretty big problem, at least big enough to warrant a minimum amount of thought and a couple of sentences in the discussion.

We agree with the general sentiment and address the Reviewer’s comment in two ways. First, we would like to point out that every statistical analysis approach will suffer from the same problem: statistical significance is not the same as biological significance. The best way to interpret statistical significance is through perturbation experiments, as we now explain in the revised “Validation of the inferred dictionaries” section. However, doing such experiments is beyond both the scope of the paper and the limit of current experimental technology, since precisely stimulating individual neurons in behaving songbirds is not yet achievable. Therefore, the field is unable to fully validate uBIA (or, really, any correlation-based analysis) using perturbations.

Second, we have added a paragraph to our discussion of the interpretability of the spiking patterns identified by uBIA. The revised “Dictionaries for exploratory vs. typical behaviors” now ends with a paragraph that includes an expanded discussion of the role that brain nucleus LMAN might play in motor exploration (per the suggestion of Reviewer 2 in their recent comments).

Reviewer #3 (Recommendations for the authors):The authors already made a great effort to improve their manuscript. However, from my side, I provide the following concerns about the significance and broad impact of the current revised manuscript.

We thank the Reviewer for these helpful suggestions.

Strength of the manuscriptThe traditional model of maximum entropy type requires a large amount of data to predict effective interactions among elements (e.g., neurons), and fails to identify statistically over- or under-represented (relative to a null model) codewords, and thus the neural-behavior prediction remains elusive. This manuscript provides a novel approach that can systematically search for the significant codewords, yet requiring fewer data samples (compared to the number of higher-order model parameters).

Thank you, we agree with this assessment.

The method first writes all orders of interactions in a probability distribution form; finding non-zero higher order interaction parameters could tell which codewords should be included in the neural dictionary and how they matter. To make the method algorithmically solvable, the authors turn the optimization into an Ising model of interacting indicator variables. This indicator variable is responsible for identification of key codewords. The inferred value of the external fields signals the importance of the codeword, while the sign of the high-order interaction parameter reflects whether the word is over or under-represented. To explain the neural-behavior correlates, the authors also include the behavior variable into the components of the activity. This framework works consistently in the songbird motor experiments.

Thank you for this correct summary.

Weakness of the manuscriptTo relate the neural activity and behavior, the authors test their method in both synthetic data and songbird real data. There are two weakness, considering the impact of the method in a broad context. First, the variable size is limited to N~20, which is severely below the number of neurons the neural data scientists get interested in (e.g., for modeling retinal ganglion cells, and even cortical cells). This may be a tradeoff between all order of interactions to be considered and the number of model parameters.

Indeed, this is a tradeoff between being systematic in detecting codewords of all orders, and the dataset sizes. Systematic analysis of codewords to all orders will necessarily require smaller *N* at the same number of recordings, *M*, compared to methods that focus on linear or, at most, pairwise codes. That said, we also point out that in many applications, specifically in recording from motor units, *N* of just a few is the state of the art – recording even from a handful of units at the same time is already hard. This and similar contexts is what our methods are devised for, as not all of biological data have become as high dimensional rapidly as, say, neuropixels. We have introduced the relevant discussion in the Discussion section of the manuscript, as well as in the synthetic data sections, where we explain why we chose to focus on datasets with specific properties.

Second, a null model is chosen to be an independent model that neglects correlations in the data, which may affect the characterization of the redundancy of the uBIA model. Even if an irreducible ensemble of codewords is obtained, what is the predictive power of this ensemble with respect to control the macroscopic behavior (e.g., skilled motor control or sensorimotor learning), rather than showing statistics of neural-behavior activity, remains obscure in the current manuscript.

These are absolutely valid concerns. However, as we pointed out above, the same questions can be asked of essentially all neural motor control papers. We are eager to publish this manuscript precisely for the reason that it’s time to leave the development of statistical methods behind us, and to focus on the biology that can be discovered by applying them to learn new biology. As we explained above, we introduced a few sentences to discuss the interpretability questions in the sections on experimental data.

Considering the strength and weakness of the manuscript, I recommend the following issues for the authors to improve their idea.1. The proposed model is limited to symmetric couplings and i.i.d data samples. First, in a neural circuit, the asymmetric coupling is common among neurons; Second, the temporal order of each spiking activity may play a key role in encoding information. For example, a paper "Blindfold learning of an accurate neural metric" (PNAS, 2018) takes the temporal information into account. When putting in the context of neural-behavior relationship, these two factors would become important. But they seem neglected in the current manuscript.

The Reviewer is absolutely correct that temporal correlations matter, that (symmetric) statistical correlations are not the same as (asymmetric) neural interactions, and so on. However, any statistical analysis method must have a well delineated domain of applicability. There are certainly a lot of scenarios where symmetric correlations without time ordering are sufficient to answer questions. Our approach focuses on those. We added these caveats to the Discussion, when we summarize pros and cons of the method. However, we chose not to reference the above mentioned PNAS article, as it is only tangentially related to our work in our opinion.

2. To achieve a higher-order-interaction-included and non-redundant model is really important problem in the field. For example, the reliable interaction model proposed in ref. 26 is not fully compared in the current manuscript, i.e., what is the new advance?

Please see section “Direct application of MaxEnt methods to synthetic and experimental data”, which deals with comparisons to the Ganmor et al. work.

This is better to clearly demonstrated in a plot in the main text.

As we have discussed above in the response to the Reviewer 2 minor comment about the structure of the paper, we strongly believe that this comparison should be kept out of the main paper text. The main reason is that, as we have argued consistently, the Ganmor approach and uBIA are not competing, but are complementary: uBIA identifies the dictionary, and the Ganmor approach can build a model on that dictionary. Since, unaided by the constraints of uBIA, the latter makes assumptions incompatible with the statistics of the data we have, it cannot work well on our data, and the above referenced section in the Methods demonstrates this beyond doubt. However, taking this as an evidence of success of uBIA over other methods would be disingenuous. As an analogy, if a Jeep gets through the mud where a Corvette gets stuck, it’s only an indication that these cars are designed for different tasks, and people would not consider a comparison of the two newsworthy. We similarly think that the comparison of uBIA and the Ganmor’s approach is not newsworthy enough to be in the main text.

Second, a recent paper (Phys Rev E, 104, 024407, 2021) used an information-based criterion to identify statistically significant couplings, which may be applicable to the context the authors consider here. In this eLife submission, the method relies on the magnetization inclusion threshold, which may be expensive for real data analysis.

This is an interesting paper (we will refer to it as the PRE method hereafter). Thank you for alerting us to it. There are substantial differences between this method and ours. Most importantly, the PRE method aims to build a generative model of the data, not just to identify which features of must be accounted for in the model. Thus, other things being equal, on very general grounds we know that it would require more data. However, other things are not equal. The PRE method is a *pairwise* methods (though the decimation discussed there may introduce some effective higher order couplings, but only of a very specific structure), and thus it cannot account for higher order structures, which uBIA notices. Finally, the PRE method detects important interactions by starting with a fully connected model, and decimating the weakest interactions turn by turn. Such approach will not work when the number of interactions, which scales as *N*^2^ increases; in contrast, we expect uBIA to be limited by the number of patterns actually seen in the data, and hence depend largely on *M* and not *N* in its complexity. We decided not to repeat this long discussion in the paper text, but we now mention this PRE article in the “Overview of prior related methods in the literature” section, and discuss the reduction in the number of interaction terms that it provides.

3. On the equation 1, the authors seem to use the local model parameter to detect the global significance of the collective codeword, which I could not understand very well, because a codeword is determined by all order of interactions considered in the log-linear model.

Equation 1 does not define the significance. It is just the all-orders model of the underlying probability distribution, and the significance is not defined till we introduce *s_µ_* and calculate their postrior expectations much later in the paper, up to Equation 16.

4. Technically, the logic going from Equation 7 to Equation 8 is broken, since the s-dependence in Equation 8 does not naturally arise from Equation7.

Thank you! This has been fixed now by re-introducing the indicator variables between Equations 7 and 8 explicitly.

5. I could not understand well how the sign of reflects whether the word is over- or under-represented, and even, how the parameter account for the reducibility of the dictionaries. This part is better to be expanded in the manuscript, although these parameters have highly non-linear function relationship.

Equation 10 shows that, to include a word or not is determined largely by the deviation of its frequency from the null model. By comparing the expected and the empirical frequencies (overbarred and angle-bracketed quantities), we know if the word is over- or underrepresented. This is described in the paragraph that starts with “Equation (10) has a straightforward interpretation.”